

# Predictive maintenance in Industry 4.0: a survey of planning models and machine learning techniques

Ida Hector and Rukmani Panjanathan

School of Computer Science and Engineering, Vellore Institute of Technology Chennai, Chennai, Tamil Nadu, India

## ABSTRACT

Equipment downtime resulting from maintenance in various sectors around the globe has become a major concern. The effectiveness of conventional reactive maintenance methods in addressing interruptions and enhancing operational efficiency has become inadequate. Therefore, acknowledging the constraints associated with reactive maintenance and the growing need for proactive approaches to proactively detect possible breakdowns is necessary. The need for optimisation of asset management and reduction of costly downtime emerges from the demand for industries. The work highlights the use of Internet of Things (IoT)-enabled Predictive Maintenance (PdM) as a revolutionary strategy across many sectors. This article presents a picture of a future in which the use of IoT technology and sophisticated analytics will enable the prediction and proactive mitigation of probable equipment failures. This literature study has great importance as it thoroughly explores the complex steps and techniques necessary for the development and implementation of efficient PdM solutions. The study offers useful insights into the optimisation of maintenance methods and the enhancement of operational efficiency by analysing current information and approaches. The article outlines essential stages in the application of PdM, encompassing underlying design factors, data preparation, feature selection, and decision modelling. Additionally, the study discusses a range of ML models and methodologies for monitoring conditions. In order to enhance maintenance plans, it is necessary to prioritise ongoing study and improvement in the field of PdM. The potential for boosting PdM skills and guaranteeing the competitiveness of companies in the global economy is significant through the incorporation of IoT, Artificial Intelligence (AI), and advanced analytics.

Corresponding author
Rukmani Panjanathan,
rukmani.p@vit.ac.in

# INTRODUCTION

Digitization of manufacturing has paved the way for industries to significantly transform the way products are produced. Industries in the current era have entered the "Fourth Industrial Revolution", also known as "Industry 4.0", where the amalgamation of real and digital systems plays a vital role. The fourth industrial revolution constructs upon the usages of computers and technology made possible by the third industrial revolution

by introducing smart and fully independent frameworks that are driven by data and AI. Beginning with the first industrialization (automation using hydro as well as thermal energy) in the age of Industry 2.0 (employing electrical energy for factory floors and large manufacturing), Industry 4.0 (I4.0) and its core technologies are crucial for enabling industrial systems to become autonomous (*Kumar, Zindani & Davim, 2019*; *Rusliyawati, Damayanti & Prawira, 2020*) and, as a result, enabling automated data gathering from industrial components and equipment. Prognostics and Health Management (PHM) is a concept that has emerged with I4.0 and has become a trend that has drawn the great attention of researchers and practitioners in the context of industrial big data and smart manufacturing. It also specifies a consistent strategy for maintaining industrial hardware's life. To achieve zero accidents, failures, or shutdowns throughout the production system, PdM can play a significant part in the prognostics of the assets. Machine learning (ML) algorithms may be used for automatic defect identification and investigation depending on the type of information that has been collected. However, choosing the right ML methodologies, data types, data sizes, and tools to use ML in industrial systems is very challenging. Time loss and impractical maintenance scheduling might result from the selection of an ineffective PdM approach, dataset, and data size. To identify appropriate ML methodologies, quantity of data, and data type to construct a usable ML solution, academicians and practitioners should consider carrying out an extensive literature survey on the existing studies and ML applications. To identify problems and classify trends to decrease or eliminate abrupt equipment failure, PdM needs a significant amount of essential data. This eliminates the reason for downtime in industrial companies. In an Industrial Internet of Things (IIoT) scenario, sensors may be used to collect the data required for PdM, which can then be analysed to find abnormalities and perceptions that would take a person a long time to notice. I4.0 gives producers the possibility to enhance their processes quickly and professionally, overcoming the difficulty of recognising the issue that needs to be fixed. It is possible to prevent failures and breakdowns by spotting abnormalities in the data collected by sensor-equipped equipment (*Ranjith, 2022*) and rectifying them promptly. This will also boost industrial productivity, which will increase yearly profit. These massive data sets amassed for ML include a plethora of relevant knowledge and data that may improve the overall efficiency of industrial processes and system dynamics. Additionally, in several scenarios, including condition-based maintenance and health monitoring, these data may be utilised to enhance decision-making (*Hwang et al., 2018*). The ability to obtain operational data and process conditions data to a significant amount has been made feasible by recent advancements in technology, information methods, computerised control, and communication networks. In order to build an automated fault detection and diagnosis (FDD) system, this kind of data is generated from various equipment parts (*Mirnaghi & Haghighat, 2020*). The statistics gathered can also be used to create more effective approaches for intelligent PdM operations. A few advantages that ML implementations provide include lower maintenance costs, fewer service pauses, fewer equipment malfunctions, longer equipment life, reduced inventories, improved engineer safety, increased productivity, service validation, higher overall profits, and many more. These benefits are also extremely and

closely related to maintenance methods (*Çınar et al., 2020*). Additionally, one of the essential elements of PdM is defect detection, which is essential for companies to do so. The following primary classes (*Munirathinam & Ramadoss, 2014*; *Rivera Torres et al., 2018*; *Jezzini et al., 2013*; *Kumar, Chinnam & Tseng, 2019*; *Mathew et al., 2017*) can be used to group maintenance policy techniques.

a. Run 2 Failure (R2F) is an unplanned maintenance that involves fixing equipment after it has broken down, leading to prolonged downtime and subpar products.
b. Scheduled maintenance also known as preventive maintenance, is applied on a schedule to prevent equipment breakdowns, but it can increase the cost.
c. Condition-based maintenance (CBM) involves continuous monitoring of equipment health and is only executed when necessary, after process deterioration.
d. PdM uses prediction tools and historical data to schedule maintenance, optimize asset use, minimize errors, and extend equipment lifespan. It is a promising technique in I4.0.

Recently, ML has emerged as one of the most potent AI techniques that can be used in a variety of applications to create smart prediction algorithms. Over the years, it has expanded into a sizable field of study. With the emergence of I4.0, there is a plethora of data generated from various sources such as IoT data, cybersecurity data, mobile data, business data, social media data, healthcare data, *etc*. ML plays a vital role for these data to be examined to build a subsequent self-regulating application for the same. Moreover, deep learning, a subset of ML also can intelligently examine enormous data (*Sarker, 2021*). ML techniques are recognised to offer several benefits as they can handle multidimensional, large feature information and uncover latent linkages across datasets in unpredictable, complicated, and challenging situations (*Bousdekis et al., 2019*; *Testik, 2011*; *Mistry et al., 2020*). However, performance and benefits may vary based on factors such as the ML technique selected, the data availability, and the problem at hand. As of now, ML approaches are often used in many fields of manufacturing, including maintenance, optimization, troubleshooting, and control (*Shafiee & Sørensen, 2019*). Therefore, the purpose of this work is to provide the current developments in ML methods used for PdM from a wide angle. This study seeks to identify and classify PdM approaches according to the ML and Deep Learning techniques. The layout and organization of the document are outlined as follows: This section first provides a quick overview of the present field of inquiry. Second, "Survey Methodology" gives the procedures for PdM planning. Thirdly, "Procedures for Planning and Implementing PdM" describes the various PdM and ML techniques used. The various applications of ML methods used in PdM are presented in "Literature Survey", followed by a comparative study of the various ML models. A summary and recommendations for further research are provided at the end.

## SURVEY METHODOLOGY

To confirm the credibility of the methodologies, much of the research in this review article is peer-reviewed publications; these investigations include conference proceedings and journal papers. Articles published from 1996 to 2023 on PdM are included in this review.

To compile the first list of publications for this research, significant sources including but not limited to Google Scholar, IEEE Xplore, Springer, ACM Digital Library, Scopus, Web of Science, and Science Direct were surveyed. Based on the abstract, certain keywords were utilised to query the databases. The referencing sections of the chosen articles were used to narrow down the pool of publications even more since they include additional pertinent research on ML and deep learning for PdM. The collected papers were segregated based on their relevance with the techniques used for PdM as papers with ML techniques used in PdM were stored and analysed separately and studies highlighting the deep learning techniques for PdM were grouped separately.

The search approach begins by utilising defined scientific databases to compile an initial compilation of articles through the implementation of a precise search string and the use of carefully defined selection criteria. Subsequently, the compilation can be expanded by including relevant works that are obtained from the first search. The research was carried out using three widely recognised scientific databases. The first search string has been established according to the key terms used for identifying PdM applications. Therefore, the string must have at least one of the following terms: PdM, condition-based monitoring, and prognostic health management. Additionally, the string needs to comprise at minimum one phrase related to the application, such as I4.0 and Smart factory. Furthermore, the string should also incorporate ML. The final string may be expressed as follows: ("PdM" OR "Condition base monitoring" OR "Prognostic Health and Management") AND ("Industry 4.0" OR "IIoT" OR "Smart Factory") AND ("Machine Learning"). Setting this as the main search string, other relevant search terms include FDD system, Planning model, data cleaning, data normalisation, FS architecture, FS techniques, filter techniques, wrapper techniques, integrated techniques, remaining useful life (RUL), anomaly identification, continuous deterioration analysis, service effects modelling, performance assessment, supervised learning, classification algorithms, regression algorithms, unsupervised learning, clustering algorithms, dimensionality reduction, semi-supervised learning, anomaly detection and fault diagnostic.

## PROCEDURES FOR PLANNING AND IMPLEMENTING PDM
### Built-in design
PdM enhances product quality while lowering downtime by forecasting system failures. Two techniques employed to gather condition data during continuous monitoring and inspection are the direct (offline) monitoring and the indirect (online) monitoring technique, where either one of the techniques can be used. The offline monitoring technique requires the maintenance team to be present physically to supervise the machinery regularly interrupting the processes. On the contrary, online monitoring, without interrupting the processes, supervises the machines constantly using sensors (*Abidi, Mohammed & Alkhalefah, 2022*). For PdM decision-making, it is further necessary to combine several types of data, including maintaining data-logs of maintenance activities, identifying causes of failures, collecting data for monitoring, analysing the impacts of failures, and setting maintenance requests for any necessary repairs or maintenance. Figure 1 depicts the PdM planning model's design.

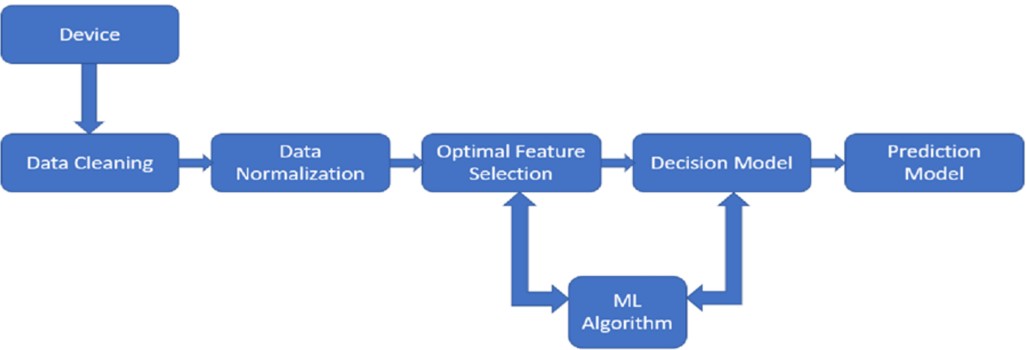

**Figure 1 The design of predictive maintenance planning model.** The PdM planning model contains five key stages: data cleansing, data normalisation, optimal feature extraction, decision model, and prediction model.

The PdM planning model contains five key stages: data cleansing, data normalisation, optimal feature extraction, decision model, and prediction model. First, the datasets are cleaned by locating misfits and adding any missing data. After being cleaned, the values are normalized, putting them inside a defined range (0–1). This is followed by the removal of redundant data by optimal feature extraction (*Sisode & Devare, 2023*). With the widespread use of sensing devices for condition tracking brought on by I4.0, making decisions under the constraints of time is now made easier. As a result, the period between the occurrence of an expected breakdown and the point at which it breaks down into an operational failure is the time during which decision-making algorithms can be used to suggest measures required to prevent or minimize the effects of the expected malfunction (*Zine, Belkhiri & Mohamed, 2022*). Though this article discusses the data normalisation and FS steps for the PdM planning model, they need not be mandatory steps for a standalone application of ML techniques for maintenance problems. They may or may not be necessary depending on the research question, the ML algorithm selected, data availability, *etc.*

## Data cleaning

When data is cleaned, problems are identified in information and fixed by referring to all different kinds of actions and operations in the dataset. Here, abnormality recognition and value replacement are used to achieve data cleansing (*Abidi, Mohammed & Alkhalefah, 2022*).

Outlier data: In analytics, outliers are values that differ notably from the other values in a dataset—either by being considerably larger or smaller. For various applications, some outlier identification models have already been presented where these techniques vary in how generic they are (*Akhiat, Chahhou & Zinedine, 2018*). Data patterns known as outliers do not correspond to the usual dynamics in the data. For example, misfits can be found and replaced by utilising the fill approach through the filloutliers() MATLAB function. To obtain the remaining components and decompose them early, anomaly detection is performed.

## Missing data management

This problem happens whenever there are certain values absent from the data. Both managing incomplete data and analysing the issues brought about by missing data need more computing and analytical time. The fillmissing() MATLAB function can replace missing records in an array with fixed values to fill in missing information (*Akhiat, Chahhou & Zinedine, 2019*). The function carries out the synthesis of assumptions and residuals. The variables may also be divided as residual and approximative components using harmonic analysis. After that, data recovery is done. The residual component is a distortion element, while the primary value is used as the approximation component. An arbitrary value is determined after calculating the standard deviation and mean of the residual component. After that, missing values are filled in using the distortions or the total of the assumption components. The algorithm might have eventually rebuilt the data. Once missing data is filled and outliers are found, the cleansed data will move to normalisation.

## Data normalisation

Data normalisation is a preliminary processing technique that transforms characteristics into a standard range, preventing larger values from dominating lesser ones. The primary goal is to reduce the bias of characteristics that have a bigger numerical impact in distinguishing pattern classes. If the relative significance of the characteristics is not known, every attribute in the data is given identical weight when determining the final class of an unexpected sample. Statistical learning methods benefit from the equal contribution of all data characteristics. Though data normalisation assures that all characteristics provide an equivalent numerical impact, this does not indicate that they are identically relevant in the classification result. Certain data aspects may be important to varied degrees, whereas others may be completely useless or superfluous (*Singh & Singh, 2020*). There are several normalisation approaches such as mean and standard deviation based normalization methods, minimum–maximum value based normalization methods, decimal scaling normalization (DSN), median and median absolute deviation normalization (MMADN), tanh based normalization (TN), and sigmoidal normalization (*Singh & Singh, 2020*).

## Optimal feature selection

Feature selection (FS) is a component of feature weighting, yet both methods vary in terms of feature significance. The existence of undesirable features hampers learning and expands the feature space. These characteristics conflict with valuable features, confusing the learning process and reducing the accuracy of classification. It also raises the processing efficiency of ML algorithms, which is determined by the number of features and occurrences in the training data. Whenever the importance of characteristics varies, FS operates superior to data weighting (*Singh & Singh, 2020*). FS looks for selecting useful features while rejecting inappropriate and superfluous ones. Whereas, feature weighting relies on the idea of feature significance, assigning weight to every feature based on its significance. Therefore, the weight for undesirable traits is expected to be zero, although it

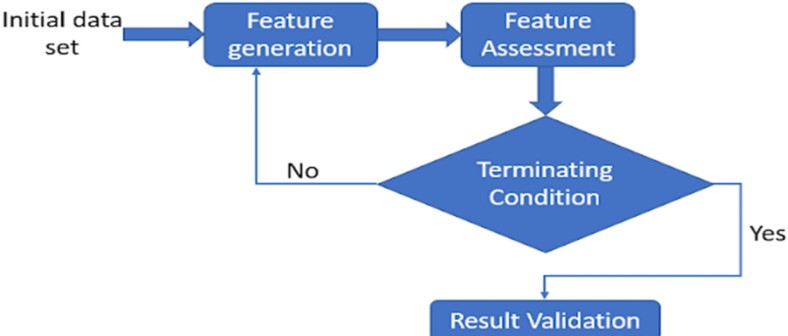

**Figure 2 Basic feature selection architecture.** A generic framework may be derived from the majority of techniques for selecting features. while feature generation, feature assessment, terminating condition, and result validation are its four fundamental processes.

may range between greater to lower for different features. As FS relies on the concept of choosing the most efficient sample within a set of features, redundant and insignificant ones are deleted from the entire collection since they impair or fail to impact the learning process. This may alternatively be considered as feature weighting, with weights assigned to binary values (*Singh & Singh, 2020*). It is a stochastic optimisation problem that requires an optimum collection of features to get accurate classification outcomes. Many approaches are currently developed for determining the optimum subset, which is divided into the following three groups: filter, wrapper, and hybrid. These approaches are discussed in detail in the following sections. A few approaches that have been developed for determining subsets of features are mutual information, value-difference metric, iterative RELIEF, cross-category feature importance, particle swarm optimisation, differential evolution, teacher learner-based optimisation, and ant colony optimisation (*Singh & Singh, 2020*). A generic framework may be derived from most techniques for selecting features. Feature generation, feature assessment, terminating condition, and result validation are its four fundamental processes (see Fig. 2) (*Dhal & Azad, 2021*). Until an ending requirement is fulfilled, the approach for choosing the variables, generate a subclass and assess it (*Xue, Yao & Wu, 2018*). The classification method then uses actual data to verify the subset that was discovered.

## Feature selection architecture

### Feature generation

A search technique also called as subset generation creates subsets of characteristics for assessment. With a low N, a thorough search over the feature space is unrealistic since there are 2N potential subsets overall, where N denotes the total variables in the underlying dataset. The subsets are frequently constructed using non-deterministic search techniques like evolutionary search (*Yang & Honavar, 1998*). Heuristic search techniques are an alternative that may be used. There are two primary families of these techniques: backward elimination and forward addition (*Koller & Sahami, 1996*), where we start using a null subclass and add features each after the other through local search.

*Feature assessment*

A specific evaluation criterion must be applied to each subset produced by the generation technique, and it must be compared to the prior subset that is better than the one given in the criteria. If better, it takes the place of the best previous subset. Consideration of the classifier algorithm's efficiency when it uses the subset in question is a straightforward way of assessing a subset. The classification algorithm is enclosed in the loop in this instance; hence the method is categorized as a wrapper. Contrarily, filter approaches utilise other factors based on association theories rather than the classifier algorithm.

*Terminating condition*

Without a sufficient criterion for terminating, the FS process can continue indefinitely before coming to an end. One of the following reasonable standards may cause the FS process to end: (1) A predetermined number of characteristics are chosen, (2) A predetermined number of iterations are completed, (3) If the addition (or deletion) of a feature does not result in a better subset, (4) The evaluation criterion is satisfied with the achieved optimal subset.

*Result validation*

It is necessary to confirm whether the optimal subset of features has been chosen. By performing various analysis on the ideal dataset and chosen subset, and also evaluating outcomes based on inaccurate data sets and/or actual data sets, it is necessary to confirm the optimal subset of features that have been chosen.

## Feature selection techniques

It is a current area of research in ML because it is a crucial pre-processing step that works well in several real-world applications. The supervised, unsupervised, and semi-supervised learning models include several types of FS algorithms, each adapted to a given application depending on the availability and nature of labelled data. There are typically three types of supervised FS techniques: filter, wrapper, and embedded approach. We will primarily concentrate on FS in high-dimensional classification challenges in this section.

*Filter techniques*

Based on the inherent properties of the data, filters assess the significance of the features. The pre-processing stage for this category is separate from the induction procedure. The filtering procedure includes two crucial steps. It initially ranks each feature according to a certain criteria measure, such as distance, Pearson correlation, and entropy, in order to create the classification model (*Roelofs et al., 2019*). Second, it uses a threshold value to choose the top-ranked characteristics. The remaining aspects are viewed as superfluous and useless. The chosen subset of characteristics is then sent into the induction classifier. Because filters are quick, they are better suited for datasets with high dimensions (*Roelofs et al., 2019*; *Yassine, Mohamed & Zinedine, 2017*; *Akhiat et al., 2020*). Repetitive characteristics may be chosen since the relationships between the independent variables are not taken into account. Two popular filtering techniques are mutual information (MI) and relief.

**Table 1 Benefits and drawbacks of feature selection approaches.** The integrated method is a hybrid of filter and wrapper techniques, in contrast to the wrapper technique, which employs a systematic analysis motivated by the findings of the classifier.

| Technique | Benefits | Drawbacks |
|---|---|---|
| Filters technique | Purely based on the learning approach | Relationships connecting attributes are disregarded |
| | Speedy execution | No correlation with the learning model |
| | Compatible with high-dimensional data | No correlation with the learning model |
| | Excellent applicability | |
| Wrappers technique | Improved accuracy obtainable | Extremely costly with respect to the turnaround times |
| | Consider how various features relate with one another | Inclined to over-fit |
| | Generate complex cognitive feature interaction | Every subset has an entirely new machine learning model |
| Integrated techniques | Quick compared to wrappers | Learning algorithm specific |
| | Reliable | Classifier-dependent selection |
| | Investigates the connection among features | |
| | Shows the relations between features | |

*Wrapper techniques*

The wrapper technique, on the other hand, builds models from scratch for each produced subset (*Akhiat et al., 2021*). Then, as a criteria function to assess its effectiveness, it employs prediction performance. This category considers how characteristics interact with one another. In general, Wrappers outperform Filters in terms of performance. However, in terms of the complexity of the models and the amount of resources needed, they are highly expensive. In the literature, a lot of wrapper strategies have been suggested (*Akhiat, Chahhou & Zinedine, 2018*; *Akhiat, Chahhou & Zinedine, 2019*; *Yassine, Mohamed & Zinedine, 2017*; *Akhiat et al., 2020*) but support vector machine recursive feature elimination (SVM-RFE) and sequential forward selection (SFS) are the most used techniques.

*Integrated techniques*

The integrated method is a hybrid of filter and wrapper techniques, in contrast to the wrapper technique, which employs a systematic analysis motivated by the findings of the classifier. Utilizing the procedure of training itself, it does feature extraction and develops an optimised classifier. With reduced computing costs than wrappers, the integrated technique, which straddles the fence between filters and wrappers, selects properties that arise during the training process depending on the classifier's evaluation standards. The most popular integrated techniques are regularisation strategies. LASSO (L1-Regularization) Least Absolute Shrinkage and Selection Operator and RIDGE (L2-Regularization) are some of the efficient approaches used. The benefits and challenges of the three techniques discussed above are described in Table 1.

## Decision modelling

Decision—making process in PdM pertains to the stage that is initiated by sensors-enabled, dynamic prediction to produce early recommendations about planned repair and methods that minimise or lessen the impact of the predicted malfunction. As an outcome

of I4.0, for PdM, which enables short-time decision-making. It is therefore feasible to consider the P-F interval, or the duration between the event occurring of a likely breakdown and the point at which it transforms into a mechanical failure, as a door of possibility in which decision-making techniques can suggest actions meant to avoid the predicted operational malfunction or decrease its impacts. A variety of approaches and strategies that attempt to improve decision-making have been developed as a result of the manufacturing environment's sophistication and unpredictable nature (*Ruschel, Santos & Loures, 2017*). Intelligent decision-making is one of I4.0's core aspects, and it is made possible by analysing information from sensors (*Sharma et al., 2021*).

Nevertheless, because of the ambiguity in predictive analytics, the deterioration operation, and the time constraints that must be met, the application of the decision-making techniques is complicated. As time has progressed, maintenance management has adopted the use of PdM as a leading method, there has been an increase in research in methods created to better assist maintenance decisions that prolong machinery operability.

### Strategy and schedule for repairs

Methods used in maintenance management and scheduling can suggest the optimal maintenance procedures in compliance with the industry's regulations and valuations of the possible negative consequences and liabilities of the prospective procedures (*Ruschel, Santos & Loures, 2017*).

### Determining decision based on reliability and degradation

To reduce protracted expenses and make it possible to schedule mitigation repair measures, this section comprises models that incorporate deterioration levels. In order to assist in analysing the balance between expense and desired quality performance, researchers may also utilise information about the machinery's current condition through the use of probabilistic approaches (*Ruschel, Santos & Loures, 2017*).

### Collective optimization

Techniques that try to improve maintenance procedures even while incorporating goals linked to productivity and the distribution network, fall under this category. Eliminating servicing expenses might not always lead to the highest possible supply, as it may also cause bottlenecks with commodity shipment and making. Instead, finding ways to manage both maintenance and other objectives can help maximize supply (*Bousdekis et al., 2021*).

### System optimization for several states and components

Techniques throughout this field enable the detection of the transitional stages of the machinery's health condition. Accordingly, analysis and design models provide intermediary decisions. Compared to studies addressing static systems, most researchers use distinct methodologies (*Bousdekis et al., 2021*).

### Analysis and management of operation costs and risks

This field contains procedures that address both pricing as well as threat estimate issues and can help with the selection of the best-planned maintenance. It is sometimes possible to predict and calculate maintenance costs for hypothetical situations or to identify

important factors. Some methods connect the consequences of failures to financial issues (*Ruschel, Santos & Loures, 2017*).

## LITERATURE SURVEY

### PdM and ML models

RUL and anomaly identification are two management tools for asset health for which the PHM system is currently a secure solution. It is accomplished through the methodical application of recent advancements in IT and AI technology (*Nelson & Culp, 2022*). Based on the outcomes of the prediction, the emerging problems that could result in catastrophic breakdowns can be rightly predicted, and suitable procedures can be made in order to prevent these flaws (*Nelson & Culp, 2022*). Industrial machinery can be changed or fixed before a flaw appears, returning the machine to its initial state until every servicing task is finished. A machine, component, system, or other piece of equipment's status can also always be evaluated. It is possible to achieve efficiency with minimal downtime by anticipating and planning for periodic outages (*Paul, Biswas & Mukherjee, 2022*). Leveraging predictive data to precisely arrange upcoming maintenance operations is the major emphasis of PdM (*Zonta et al., 2022*). The PdM modelling process essentially entails four interconnected stages: (i) continuous deterioration analysis; (ii) service effects modelling; (iii) maintenance policies formulation; and (iv) performance assessment.

### *Continuous deterioration analysis*

For system health prognostics, it is crucial. It largely relies on Lévy or diffusion stochastic processes which align with condition monitoring data in the current state of the art. The Gamma function is currently the most popular option for linear deterioration due to its physical significance and versatility in mathematics. If this option is unsuccessful (as in the example of data on the deterioration of GaAs lasers (*Duan & Wang, 2022*) or energy pipeline corrosion (*Liu & Bao, 2022*)), the inverse Gaussian (IG) technique may be a viable fallback. While still having the same conceptual and mathematical implications as the Gamma process, it is demonstrated to be more versatile in adding stochastic impacts and determinants (*Lindgren, Bolin & Rue, 2022*). This clarifies why the IG process has lately gained a lot of attention, particularly in the optimization of accelerated degradation tests (*Lipu et al., 2022*) and in the assessment of the system's RUL (*Li et al., 2022*). Despite its adaptability, its applications in PdM modelling are still quite rare.

### *Service effects modelling*

This paradigm explains how maintenance operations affect how the equipment degrades. The best-known assumption in the literature is unquestionably perfect maintenance with the as-good-as-new (AGAN) effect (*Cortés Olivares, 2022*). It cannot, however, account for many practical activities whose flaws may result from influencing factors like a human mistake, the calibre of the replacement parts, a shortage of supplies, a limitation of service time, *etc*. Imperfect maintenance (IM) models have been widely researched to address such realistic objectives under the presumption that the system condition following a service is worse-than-new but better-than-old. Regardless of these measures, predicting IM effects concerning degradation processes is largely inactive and

mostly replicates the concepts of lifetime-based IM models (see *e.g.*, *Si et al., 2022*; *Zhou et al., 2020*; *Huynh, 2021*; *Hu et al., 2021*). Additionally, the memory assumption links historical and current IM operations in most current algorithms. Because degradation processes are unpredictable, it is difficult to test this hypothesis in real-world settings. It is still unclear how to overcome this significant supposition and arrive at workable past-dependent IM models.

### Maintenance policies formulation

This framework describes the steps that should be taken at decision periods based on the changes in the device's condition. The development of this type of strategy has mostly centred on inspection schedules and benchmarks for decision-making on condition monitoring in the research (*Zheng et al., 2022*). Maybe the easiest strategy is one that has regular monitoring and a pre-set degradation-based criterion (see *e.g.*, *Truong-Ba et al., 2021*). This reduction makes implementation simple but also suggests a poor maintenance strategy. As a result, there has been a lot of focus on improving efficiency by making the policy more flexible. By adjusting the inter-inspection frequencies of the equipment aging, system deterioration grade, system degradability, device RUL, or the work atmosphere, some writers proposed aperiodic inspection techniques. Others suggested utilising various routine maintenance standards (see *e.g.*, *Zhou & Yin, 2019*). Given the large quantity of decision parameters employed by these systems, maintenance strategies are typically quite complicated and sometimes even impractical. Therefore, creating a dynamic PdM strategy that is both less complex for real-world implementations and clever enough to meet specific performance criteria is a major issue (*e.g.*, cost, availability, *etc.*).

### Performance assessment

The objective of measuring efficiency is to maximise maintenance efficiency by enhancing service strategies' decision factors (inter-inspection frequencies, service criteria, *etc.*). There are two primary methods for efficiency evaluation analytic methods and statistical models (*Barykin et al., 2021*). The first method makes use of tools for heuristic methods or strategy iteration as well as the (semi)-Markov decision process. It necessitates converting the corresponding discrete-state space from the continuous-state space of deterioration processes. Be aware, nonetheless, that in certain circumstances, when the inherent persistence of deterioration operations is of higher significance for analysis and decision-making, such a notation may be detrimental. The second strategy enables getting around this barrier. In fact, this method may be successfully used for deterioration processes having continuous-state space (*Huynh, 2021*) in addition to discrete-state space utilising the (partial) regenerative theory or (Markov) renewal theory. The full analytical cost models it produces are more significant than the exact solution the first technique produced. Lately, many researchers have evaluated and optimised the unscheduled downtime total cost of deterministic maintenance policies using the semi-regenerative approach. Further mathematical work is needed to properly formulate this approach when used to adaptive PdM strategies.

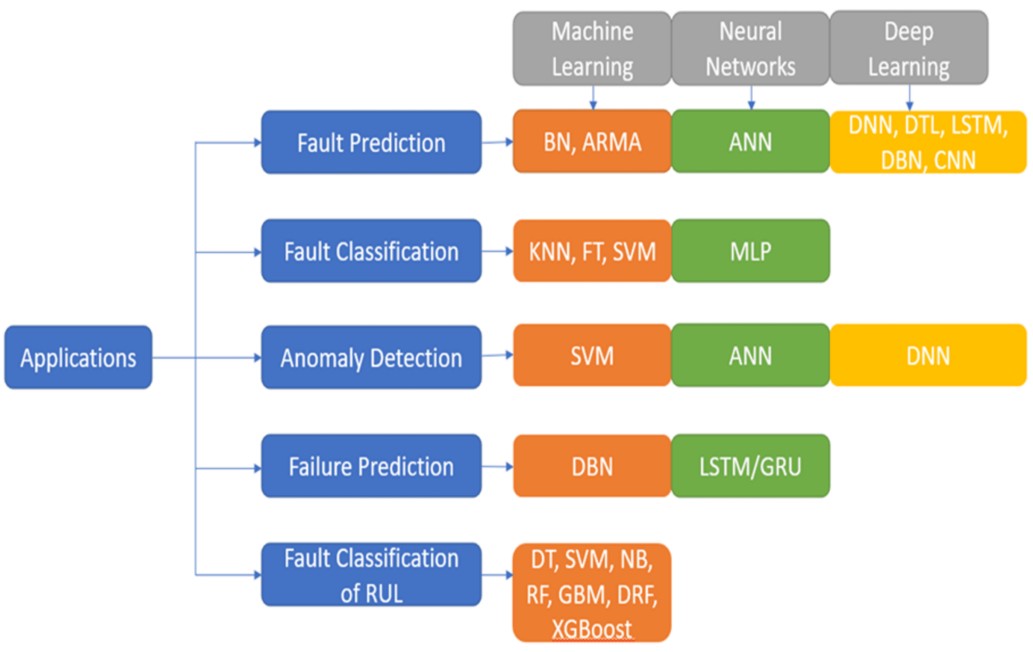

**Figure 3 Artificial intelligence approaches.** ML methods have been extensively used in many different academic domains. Choosing the best option that is also the simplest and most effective may be quite important.

## Imperative AI algorithms for PdM

AI algorithms play a key role in the prognostics of the equipment in a manufacturing I4.0. The vast amount of data created by various IoT devices in the industry can be utilised to analyse and gain meaningful insights on the RUL of equipment, anomalies identified, strategies to avoid unplanned downtime, reduce planned downtime, how maintenance resources can be optimised, and many more. Figure 3 depicts a few of the ML models and deep learning models. The models are segregated based on i) supervised learning ii) unsupervised learning iii) semi-supervised learning. The models are further classified as Classification and Regression models of supervised learning, Clustering and Dimensionality Reduction of unsupervised learning, and semi-supervised techniques. In recent years, ML methods have been extensively used in many different academic domains. Choosing an appropriate and simple algorithm that is effective is vital. Massive volumes of data from breakdown circumstances and health conditions instances must often be collected for training the ML systems. In recent years, ML and deep learning techniques have been vital tools for researchers to work extensively on PdM.

### ML algorithms in PdM

ML comprises a collection of structures and techniques for extracting meaning from data. ML is an area of AI that empowers computers to acquire knowledge from data, detect patterns, and generate outcomes with little assistance from humans. The process entails developing algorithms to execute classification, prediction, and recommendation functions using input data and datasets. The efficacy of ML is contingent upon several factors,

including data quality, algorithm selection, and the capacity to account for bias and variance to prevent overfitting. Algorithms enhance their precision as they gain knowledge from data, rendering ML a potent instrument for an extensive array of applications, including predictive analytics and image recognition. This rapidly changing domain consistently presents inventive resolutions to intricate challenges that span various sectors.

*Supervised ML algorithms*
Every instance of a dataset is seen by ML algorithms as an assortment of features. These characteristics might be continuous, categorical, or binary. This kind of learning is known as supervised learning if the examples are labelled. In supervised learning, the model is tested on unlabelled data after being trained on labelled data. The gathering of the dataset is the first step in its basic design. After that, the dataset is divided into training and testing data, and the data is pre-processed. The model is trained to identify the characteristics connected to each label by feeding the extracted features into an algorithm. In conclusion, the test data is fed into the model, which uses the anticipated labels to forecast the test data. Regression and classification constitute the two main categories of supervised learning. The next sections include a detailed discussion of both.

*Regression*
Regression is one of the supervised learning techniques in ML that predicts a continuous output variable based on input features. The input features are also referred to as independent variables and the output variables are known as dependent variables. Regression aims to identify a mathematical representation that defines the association connecting the features and the target variable (*Sarker, 2021*). Financial prediction, cost assessment, pattern identification, trend assessment, marketing, time series inference, and many more are a few of the applications where regression models are used extensively.

    **Linear regression:** It is a basic ML approach used to identify and predict the association between a dependent variable Y (expected output) and one or more independent variables X (also known as regression line) which are the features that influence the output. It looks for the best-fit linear equation under the assumption that the association is linear. Finding variables in a linear regression that minimize the gap between the values predicted by the model and the actual values in the training data, is the main objective of this model (*Ingole et al., 2022*). It is defined by the following equations:

$$y = a + bx + e \tag{1}$$
$$y = a + b_1x_1 + b_2x_2 + \cdots + b_nx_n + e, \tag{2}$$

where "e" is the error term, "b" is the line's slope, and "a" is the intercept. The slope "b" indicates the variation in Y for a unit variation in X, while the intercept "a" indicates the value of Y when X is zero. The divergence between the expected and actual values of Y is represented by the error term "e". Based on the specified predictor variable(s), the best-fit line may be used to identify the probable outcome of the target variable. Simple and Multiple Linear Regression can determine the value of Y for any given value of X using this equation. By incorporating two or more predictor variables (X) to model a response variable (Y) as a linear function, we may use the approach known as multiple linear

regression. It is an advancement of Simple Linear Regression, which models the response variable using only one independent variable. Usually, a cost function like Mean Absolute Error (MAE) or Mean Squared Error (MSE) is used to measure it. There are two primary types of linear regression: simple linear regression, which works with only one independent variable, and multiple linear regression, which involves two or more independent variables (*Sarker, 2021*).

**Lasso and Ridge regression:** Methods like Lasso and Ridge regression serve to create models with an extensive amount of data. They minimize the model's sophistication and contribute to preventing overfitting. L1 regularization, a method used in Lasso regression, reduces the size of coefficients and has the potential to completely omit a few of them. This aids in determining the model's most significant features. L2 regularization, which is used in Ridge regression, likewise reduces the overall size of the coefficients without omitting them completely. It plays a key role when multicollinearity in the data is observed, that is, when some predictors correlate with one another (*Ingole et al., 2022*).

**Elastic-Net regression:** It is a model that incorporates the principles of linear regression's L1 regularization (Lasso) and L2 regularization (Ridge). The Elastic-Net regression model plays a vital role in cases where the regression problem aims at predicting a continuous target variable using one or more independent characteristics. The model provides a dynamic technique for FS and regularization, which helps to mitigate most of the shortcomings of Lasso and Ridge regression. The linear regression model in Elastic-Net aims to minimize the cost function that is comprised of a regularization term and the mean squared error of predictions (*Münch et al., 2021*). L1 and L2 regularization are combined in the regularization term:

L1 regularization (Lasso): With L1 regularization, a penalty term is added to correspond with the regression coefficients' absolute values. Reducing certain coefficients to absolute zero, it increases sparsity in the model and thereby carries out feature selection.

L2 regularization (Ridge): The squared value of the regression coefficients is used to determine whether a penalty term in L2 regularization should be added. By largely minimizing the overall coefficients but not completely zero, the L2 regularization helps avoid overfitting.

The trade-off between L1 and L2 regularization can be controlled by adjusting the two hyperparameters, alpha and l1_ratio of elastic-net regression.

Alpha ($\alpha$): The strength of the regularization is largely defined by this hyperparameter. The $\alpha$ when set to 0, there is no regularization, and the model is similar to an ordinary least squares regression whereas when $\alpha$ is set to 1, both L1 and L2 regularization are combined, and the average value of $\alpha$ allows different levels of regularization. In other words, $\alpha$ determines the extent to which the model is penalized for having large coefficients, thus varying $\alpha$ values can result in varying levels of regularization.

L1 ratio (l1_ratio): The ratio of L1 to L2 regularization is managed by this parameter. Pure L2 regularization (Ridge) is denoted by a value of 0, while pure L1 regularization (Lasso) is denoted by a value of 1. A combination of both is possible with intermediate values.

## Classification

Assigning input data points to predetermined groups or classes is the task of classification in supervised ML. Using a labelled dataset as a training set, where each data point is assigned to a predefined class, the classification uses this trained model to estimate the class of newly collected, unlabelled data. The model is trained using the labelled dataset, and it gains the ability to recognize features in the data that correspond to each class. By applying the learnt patterns to the new data, the trained model may be used to identify the class of newly acquired, unlabelled data. A classification model's performance is usually assessed using measures like F1-score, recall, accuracy, and precision (*Kunselman et al., 2020*).

**Logistic regression:** Each independent variable's statistical significance in relation to probability is determined using this model. This method is effective for simulating binomial outcomes. For instance, using one or more explanatory factors, determine whether a person will get cancer or not by assigning values between 0 and 1. In logistic regression, the result variable has two possible values. It is a classification technique that primarily uses probability to assign observations to discrete classes (*Shah et al., 2020*). According to *Jurafsky & Martin (2021)*, logistic regression is one of the most important analytic tools in the social and natural sciences. The article also explains that logistic regression can be used to classify an observation into one of two classes (like 'positive sentiment' and 'negative sentiment'), or one of many classes. The work in *Zou et al. (2019)* proposes a new method for optimizing the logistic regression model by using a genetic algorithm. In *Sun & Dong (2020)* a new method for improving the accuracy of the logistic regression model by using a hybrid approach of particle swarm optimization and artificial bee colony algorithm is proposed. *Sheikh, Goel & Kumar (2020)* proposed a new method for predicting the probability of loan approval using logistic regression. These papers provide a comprehensive understanding of logistic regression and its applications.

**K-nearest neighbours (KNN):** The KNN algorithm is an ML technique frequently applied to regression and classification problems. KNN retains every training example that is readily accessible in memory throughout the training phase. The technique determines the distance between a newly discovered data point and all training data points predicted. Following the identification of a set of K-nearest neighbours based on the shortest distances, the method predicts by either classifying the neighbours according to the most prevalent class label or regressing the data by averaging the target values. The choice of the hyperparameter K, which denotes the number of neighbours to take into account, is a crucial step in KNN. A small K could contribute to overfitting since it increases an algorithm's sensitivity to noise and local deviations. However, a high K might cause over smoothing, in such a case the algorithm would fail to recognize minute correlations in the data. The algorithm's effectiveness is also influenced by the distance metric selected, which is often Euclidean distance. Normalizing the features guarantees that every feature contributes equally to the analysis. The proposed model in *Rajaguru & Sannasi Chakravarthy (2019)* uses decision tree and KNN algorithms for the grouping of breast tumour. The tumour can be classified as either benign or malignant. The two algorithms are tested with the Wisconsin Diagnostic Breast Cancer (WDBC) dataset where principal

component analysis (PCA) is used for feature selection. Both the ML approaches are compared by the standard performance metrics and the evaluation outcomes reveal that the performance of the KNN classifier shows better results than that of the decision-tree classifier. In *Ali et al. (2020)* the KNN method is used to tackle the classification issue. It focuses on methods for classifying neighbours using the nearest neighbour approach, which includes processes for determining distance and similarity, computational challenges in locating nearest neighbours, and data reduction strategies. Inherent dimensionality and similarity metrics for time-series are also discussed in the study.

**Naive Bayes:** The algorithm is a probabilistic classification technique that relies on the Bayes theorem to determine the likelihood of an occurrence based on past knowledge of potential confounding variables. In a number of real-world applications, naive Bayes has shown to be successful regardless of its "naive" assumption of feature independence. The Bayesian probability computation is made simpler by the technique's initial assumption that features are conditionally independent provided the classification is done. Considering the measured frequency of features and instances of them within each class, prior probabilities and chances are computed during training. The approach calculates variables like mean and standard deviation for all classes and applies a probability distribution, usually Gaussian, for continuous features. During the modelling phase, the method uses the Bayes theorem to compute the posterior probabilities for all classes providing a new data point containing features. It then guesses the class with the strongest probability. The naive Bayes method is reviewed in *Ali et al. (2020)* where the purpose of this study is to present a thorough grasp of naive Bayes and its applications in a variety of fields. It is a popular probabilistic classifier for data mining and ML applications. It has been effectively used in text categorization, spam filtering, sentiment analysis, and other fields. It is predicated on the idea that characteristics are independent of one another. The study examines many naive Bayes variants, including multinomial naive Bayes, Bernoulli naive Bayes, and Gaussian naive Bayes, emphasising their advantages and disadvantages. It also addresses the algorithm's possible weaknesses, including how sensitive it is to feature independence presumptions and how unbalanced datasets affect it.

**Linear discriminant analysis (LDA):** Another well-known method for reducing dimensionality in pre-processing steps for data analysis and ML purposes is LDA (*Obaid, Dheyab & Sabry, 2019*). Projecting a dataset is the primary goal of LDA which has a large feature count in a low-dimensional space that exhibits strong class separability. As a result, computing expenses will go down. The methodology employed by LDA bears a strong resemblance to PCA. In addition to maximising data variance (PCA), LDA maximises the separation of different classes. Projecting a dimension space onto a smaller subspace without affecting the class data is the aim of linear discriminant analysis. Other dimensionality-reduction-focused approaches have recently been presented in *Yang & Wu (2014)*, *Yu & Yang (2001)* and *Paliwal & Sharma (2012)* for investigating the localization of high-dimensional data. Nevertheless, traditional LDA is only effective if all data classes are roughly Gaussian distributed. Stated differently, multimodal data is more complicated than data with a Gaussian distribution, and LDA is not equipped to handle it. Consequently, a major obstacle faced by many dimensionality reduction techniques now

in use that rely on LDA is precisely identifying the local structure inside multimodal data. The work in *Liu et al. (2018)* uses acoustic emission signals with LDA to enhance defect identification in the Fused Deposition Modelling (FDM) process. Through the identification of latent themes or patterns in, LDA seeks to accomplish accurate defect detection. The researchers can better diagnose faults by classifying the acoustic emission signals into several fault categories by employing LDA, which allows them to extract pertinent information from the signals. The application of LDA in this study yielded findings that indicate improved fault identification in the FDM process, with the suggested methodology achieving greater accuracy in fault diagnosis when compared to other approaches.

*Unsupervised ML algorithms in PdM*

Analysing raw datasets is made easier by unsupervised ML approaches, which aid in the production of analytical insights from unlabelled data. Recent developments in factor analysis, latent models, hierarchical learning, clustering methods, and outlier identification have greatly advanced the state of the art in unsupervised ML. Recent developments in unsupervised ML, such as the creation of "deep learning" approaches have improved the state of ML considerably by making it easier to analyse raw data without necessitating meticulous engineering and domain knowledge for feature creation. We will discuss some popular unsupervised learning methods in the following section.

## Clustering

Data analysis constitutes an unsupervised learning process that extracts observations and concealed associations from data. Data analysis is increasingly important in data science and ML since it reveals all concealed information and facilitates the determination of various characteristics of the data collection. Clustering is a data analysis approach that groups data according to similarity measurements. A cluster consists of comparable items or data points. Likewise, objects or data points in different clusters will be identical. However, when these clusters are examined together, they show significant differences. Clustering is regarded as the most essential unsupervised learning approach since it involves identifying patterns in a set of unlabelled data (*Ali et al., 2020*).

**Density-based spatial clustering (DBSCAN):** An approach for grouping spatial datasets with different densities and shapes is called DBSCAN. Depending on the density of their neighbourhoods, DBSCAN divides data points into three categories: noise, border points, and core points. Clusters are formed by repetitively growing from core points, which have enough density of neighbours along a certain radius cluster (*Yang et al., 2022*). Although noise points do not constitute any cluster, border points though located close to core points are also not considered as core points. As part of the algorithm's operation, the seed point is initialized, core points are found, and clusters are expanded recursively. The algorithm's sensitivity may be adjusted by users using the parameters Eps and MinPts (a user-defined threshold). DBSCAN is good at finding clusters of any shape, resilient to outliers, and does not need to provide the number of clusters in advance. Nevertheless, it may have trouble processing high-dimensional data or clusters with different densities,

and its ability to perform might be dependent on parameter selections. Many academics have worked to enhance DBSCAN (*Wang et al., 2019*) throughout time to use it more successfully. They have done this by using meta-heuristic algorithms (*Jian & Zhu, 2021*; *Agarwal, Mehta & Abraham, 2021*; *Zhang et al., 2021*; *Singh, Singh & Kaur, 2021*) to ensure that the EPS and MinPts parameters in DBSCAN are searched and chosen automatically. An approach for multi-segment optimisation was presented in *Lai et al. (2019)*. It can swiftly pick the right EPS parameter and has a strong optimisation potential as an individual variable updating technique. It can also acquire good DBSCAN accuracy. In order to address the clustering problem, the authors in *Jian, Li & Yu (2021)* suggested an adaptive DBSCAN. This is done by using the target solution and its motion range as noise locations, where the neighbourhood is impacted by physical elements. By using the harmony search optimisation technique on DBSCAN, in *Zhu, Tang & Elahi (2021)* researcher was able to improve both the clustering parameters and the clustering outcomes. KR-DBSCAN, a density-based clustering technique employing space and reverse closest neighbour, was introduced in *Hu et al. (2021)*.

**Hierarchical agglomerative clustering (HAC):** One of the popular techniques for unsupervised learning is HAC. The performance of HAC is significantly impacted by the linking criterion that this approach uses to determine how similar both clusters are. It is a popular bottom-up clustering technique that begins by creating an initial segmentation and unites the most identical clusters until a single cluster comprising every item is created. The appropriate use of HAC is seen when the number of clusters is unknown. It generates a dendrogram that illustrates the relationships between the clusters (*Randriamihamison, Vialaneix & Neuvial, 2021*). The primary flaw in HAC is the linkage criteria, which specifies how the relationship between clusters is determined based on the separations between pairs of components. Single, average, and complete linkage are the three primary linkage criterions in HAC. Each criterion has unique properties and often yields segments with varying attributes. Analysing the distance between two clusters' closest members, single linkage calculates the degree of similarity the two clusters are to one another. It is straightforward and computationally effective; however, a chaining issue may cause extended clusters. Although average linkage provides a more intricate definition of similarity, it calculates the average proximity between every pair of points between each cluster to estimate their resemblance (*Emmendorfer & de Paula Canuto, 2021*). Finally, complete linkage is more sensitive to noise and outliers in the data as it determines how similar two clusters are based on how far apart their most distant members are from one another. Numerous studies have been conducted on hierarchical agglomerative clustering. The work in *Jianfu, Jianshuang & Huaiqing (2011)* offers the HACNJ method, a hierarchical clustering algorithm based on the Q-criterion. When working with big datasets, HACNJ's time complexity of $O(n3)$ and space complexity of $O(n2)$ is prohibitive, like that of simple hierarchical clustering. A thorough analysis of a few enhanced hierarchical clustering techniques is provided in *Rani & Rohil (2013)*. The goal of these methods is to get over the drawbacks of simple hierarchical clustering techniques. Clustering Using REpresentatives (CURE), Balanced Iterative Reducing and Clustering using Hierarchies (BIRCH), ROCK (RObust Clustering using linKs), CHEMELEON

Algorithm, Linkage techniques, Leaders—Subleaders Leaders are some of the techniques that are described.

**K-means:** This unsupervised clustering technique divides a dataset into clusters according to similarities. The total squared distance between every data point and its designated centroid is what it attempts to minimize. K-means functions exceptionally well on massive datasets and has a fast clustering speed, but it has low clustering consistency and is prone to solitary data and noise. The k-means clustering algorithm has been extensively researched in the literature, with several adaptations, and utilised in numerous academic domains (*Alhawarat & Hegazi, 2018*; *Meng et al., 2018*; *Lv et al., 2019*; *Zhu et al., 2019*). Nevertheless, initializations often have an impact on these k-means clustering methods, thus a predetermined number of clusters must be provided. Validity indicators can be applied in this situation to determine a cluster number which is expected to be unhindered by clustering techniques (*Halkidi, Batistakis & Vazirgiannis, 2001*). The literature identified a number of cluster validity indicators for the k-means clustering technique, including the Bayesian information criterion (BIC) (*Kass & Raftery, 1995*), the Akaike information criterion (AIC) (*Bozdogan, 1987*), Dunn's index (*Dunn, 1973*), the Davies-Bouldin index (DB) (*Davies & Bouldin, 1979*), the Silhouette Width (SW) (*Rousseeuw, 1987*), and the Calinski and Harabasz index (CH) (*Calinski & Harabasz, 1974*), Gap Statistic (*Tibshirani, Walther & Hastie, 2001*), generalized Dunn's index (DNg) (*Pal & Biswas, 1997*), and modified Dunn's index (DNs) (*Nejc, 2012*). The two main kinds of clustering validity indicators are internal indicators and external indicators (*Rendón et al., 2011*). Through analysing the cluster memberships generated by the clustering procedure with already available information, such as an explicitly provided class label, external indicators are applied to analyse clustering outcomes (*Lei et al., 2017*; *Wu et al., 2009*). Internal indicators, on the other hand, assess the quality of cluster topology by emphasizing the inherent information included in the data (*Jegatha Deborah, Baskaran & Kannan, 2010*). The most prevalent and often utilised approach for determining the number of clusters is X-means (*Pelleg & Moore, 2000*). This approach has been employed in previous research by authors like *Witten et al. (2000)* and *Guo et al. (2017)*. *Pelleg & Moore (2000)* expanded k-means in X-means by dividing itself and locally deciding cluster centres in every iteration of k-means to achieve improved clustering. After which, a range of cluster numbers within which the genuine cluster number falls is indicated. The splitting procedure is then carried out using a cluster validity indicator, such as BIC.

**Fuzzy C-means (FCM):** A robust unsupervised technique for data analysis and developing models is fuzzy clustering. FCM algorithm (*Ruspini, Bezdek & Keller, 2019*), which is a fuzzy clustering technique, is a very effective algorithm for processing data and generating rules from a dataset that has a high frequency of fuzzy features. Fuzzy clustering can seem more intuitive than hard clustering in a variety of scenarios. Instead of being required to completely conform to one of the classes, objects that are on the borders of many classes are given membership degrees between 0 and 1, which indicate their selective membership. The most commonly adopted approach is FCM. In 1974, (*Ruspini, Bezdek & Keller, 2019*) published the first study on FCM clustering in the literature for a particular

example (m = 2). Fuzzy partitioning is used by the FCM to allow a data unit to be assigned to any cluster with a membership grade that ranges from 0 to 1. This method uses the distance from the cluster centre to the data point as the factor to determine membership for each data point that corresponds to that cluster centre. The closer the data is to the cluster centre, the more affiliated it is with that specific cluster centre. The total membership of all the data points must correspond to one. Certain parts of data in real-world issues could be incomplete or contain ambiguous information. Fuzzy set theory, in conjunction with the membership function notion defined in the interval [0, 1], offers a suitable means of expressing and handling these kinds of data components. Every component in this concept has the potential to be associated to several clusters. In an effort to get around some of the algorithm's drawbacks and enhance its clustering capabilities in various scenarios, a number of novel techniques built on the FCM algorithm were released. A summary of these new approaches' performance may be found in *Jie et al. (2017)*.

## Dimensionality reduction

The process of converting high-dimensional data into lower dimensionality requires dimensionality reduction (*Ayesha, Hanif & Talib, 2020*). Many dimensionality reduction approaches have been applied in the recent couple of years to sort the dataset under considered data samples. In order to reduce dimensionality, high dimensionality input must be transferred to lower dimensionality inputs, resulting in the visualisation of comparable points in the input field to nearby points on the main field. The benefits of using this approach on a given dataset are reduced data storage capacity because of the reduced number of dimensions, increased computation speed, elimination of redundant, noisy, and insignificant data, enhanced data quality, increase in the accuracy and efficiency of an algorithm, and provide data visualization. In general, there are two primary categories of dimensionality reduction techniques: feature extraction (FE) and FS. FS is regarded as a crucial technique because data is generated at a rapid pace. By using FS, certain major dimensionality issues can be reduced, including redundancy and insufficient information removal. The process of feature engineering is a crucial pre-processing stage that aids in the extraction of modified features from the raw data, which serves to enhance the quality of a ML algorithm's output and simplify the ML model. FE also tackles the problem of identifying the most distinct, useful, and concise collection of features to enhance the effectiveness of data handling and storage.

**Principal component analysis** (**PCA**): While there are other dimensionality reduction solutions, PCA is the most implemented. Similar to wavelet decomposition or Fourier analysis, this provides the advantage of evaluating results between datasets and determining the relative significance of the features. It is one of the unsupervised learning methods that lower the dimensionality of data. Feature reduction from enormous data sets into smaller features that hold significant information is a common application of this technology. It streamlines complicated datasets, increases data interpretation, and boosts computing speed. By dividing the data into low-rank and sparse components, robust PCA allows it to manage noise and outliers in the data. Additionally, it can adjust to flowing data using live PCA, enabling adaptive dimensionality reduction and real-time analysis.

Exploratory data analysis is aided by PCA's ability to visualise high-dimensional data in a lower-dimensional environment. Furthermore, PCA can recognise and resolve multicollinearity problems in datasets, yielding a more autonomous component set. Nevertheless, PCA needs interpretation to determine the optimum number of components to maintain, presupposes linearity, which may result in loss of data. PCA and linear discriminant analysis (LDA), were examined in *Reddy et al. (2020)*, where the authors implemented it on the dataset available in the UCI ML repository. Features were scaled down from 36 dependent components to 26 by preserving 95% of the dataset. It was noted that PCA classifiers outperform LDA classifiers. In *Saraswathi & Gupta (2019)*, using a random forest (RF) classifier, the authors developed a multi-class tumour diagnosis strategy. They also gave a comparative study of the RF-PCA, RF, and RF-PCA approaches with random selection strategies. The experimental findings demonstrate that RF-PCA's random selection strategy yields higher accuracy compared to alternative approaches. Furthermore, in *Jamal et al. (2018)* the authors have concentrated on how feature extraction might lower the quantity of characteristics needed for the categorization of breast cancer from the initial White Blood Cell (WBC) data set. The dimensionality reduction achieved with the K-means cluster is nearly as effective as PCA, according to the metric assessment. However, in order to eliminate unnecessary features and keep the best attribute subset, *Salo, Nassif & Essex (2019)* suggests a unique hybrid approach that combines PCA with Information Gain (IG). PCA efficiency was compared with Isomap, a deep autoencoder, and a variational autoencoder. The robustness of the suggested technique produced encouraging results in both NSL-KDD and Kyoto 2006+ datasets. MNIST, Fashion-MNIST, and CIFAR-10 are three popular image datasets that were used in the experiments. Compared to its neural network counterparts, PCA's computation time was a couple of orders of magnitude quicker. A suitably wide dimension was possessed by the two auto-encoders (*Fournier & Aloise, 2019*).

**T-distributed stochastic neighbour embedding (t-SNE):** t-SNE, created by *Hinton & Roweis (2003)*, is one of the nonlinear dimension reduction techniques that has obtained a lot of research interest lately. It has been used for a variety of application areas, including remote sensing images, computational fluid dynamics, single-cell RNA-sequencing data, microbiome data, and computational fluid dynamics. The primary goal of this technique is to create probability distributions from paired distances in which greater distances are associated with lower probabilities and vice versa. This is the most popular learning technique for single-cell analysis. Multidimensional data sets might be effectively projected onto a 2D or 3D plane using the t-SNE technique, maintaining the majority of the local architecture of the data in the original high-dimensional environment. Nevertheless, the t-SNE approach is rarely used for classification or regression tasks as it lacks an integrated mechanism to translate incoming data points to the appropriate low-dimensional form *Maaten (2009)*. By implementing neural networks for feature extraction and subsequently performing classification over the projected low-dimensional space from t-SNE, several researchers have taken steps to address the out-of-sample extension challenge (*Maaten, 2009*; *Oliveira, Machad & Andrade, 2018*). For t-SNE to operate well, an array of parameters must be adjusted. One such parameter is the perplexity per, which is a smooth

measure of the effective number of neighbours. According to certain research, this parameter often has a value ranging from 5 and 50 (*Van der Maaten & Hinton, 2008*). In actuality, accurate tuning of per necessitates practical knowledge and an understanding of the internal workings of the t-SNE approach.

**Autoencoders:** Autoencoder is an artificial neural network, through unsupervised learning, can acquire new information encodings (*Charte et al., 2018*). Through a number of hidden layers, AEs are taught to acquire an internal representation that enables replication of the input into the output. In order to use the original information in the hidden layers for other activities, it is intended to be produced in a more compact representation. Due to their qualities and effectiveness, AEs are commonly employed for dimensionality reduction tasks (*Rosenberg, Hebert & Schneiderman, 2005*; *Dópido et al., 2013*; *Leistner et al., 2009*; *Lee, 2013*). An AE's fundamental structure is identical to a multilayer perceptron's. In AE the data constantly flows in one direction as it functions as a feedforward neural network with no cycles. A sequence of layers, comprising an input layer, many hidden layers, and an output layer—each layer's units coupled to those in the subsequent layer—usually make up an AE. Since the primary objective of AEs is to replicate the input into the output at every step of the learning process, the output and input both have an identical array of nodes (*Charte et al., 2018*). The encoder and the decoder are the two parts that make up the AE. The input layer and the first segment of the hidden layer constitute the first portion, while the output layer and the other half of the hidden layer form the second. It is possible to distinguish between two types of AEs based on the quantity of units present in the concealed layers. Those with fewer units in their hidden layer than in their input and output layers are referred to as undercomplete AE. Its primary goal is to have the network learn a concise form of the input to derive new, more advanced features. Those with more units in their hidden layer than in their input and output layers are referred as overcomplete AE. When obtaining an expanded form of the input, it is required to employ additional methods to avoid a major issue where the network duplicates the input to the output without learning anything beneficial. Furthermore, AEs are a great instrument for creating a new input space that is lower dimensional and composed of features at a higher level.

## Semi supervised ML algorithms in PdM

Previously, there were two main areas of ML: supervised and unsupervised learning. In supervised learning, a series of data points having an input value and an output value are shown to the learner from which a classifier or regressor model is built that is capable of estimating the output value for a set of inputs that haven't been seen before. On the contrary, no precise output value is given in unsupervised learning. Rather, an attempt is made to deduce a basic pattern from the data. The aim of unsupervised clustering is to determine an association between the supplied inputs to categories so that comparable inputs correspond to the identical category. A subfield of ML called semi-supervised learning seeks to integrate these two objectives. Semi-supervised learning algorithms often use data that usually corresponds with one of these two areas in an effort to increase efficiency in that area. For example, more data points for which the label is uncertain may

be utilised to help with the classification process while solving a classification issue. However, for clustering algorithms, the fact that some data points are part of the same class may help the learning process. The main objective of semi-supervised learning is to leverage unlabelled data to create more effective learning processes. It turns out that this isn't always feasible nor is it easy. Unlabelled data is only valuable if it contains information valuable for label prediction which is either absent from the labelled data or difficult to derive from it. The algorithm then has to be capable to retrieve this data in order to for any semi-supervised learning approaches to be implemented.

**Self-training:** Self-training models are a subset of semi-supervised learning techniques that train repeatedly with labelled and pseudo-labelled data using several supervised classifiers. Unlabelled data points that have labels applied to them by the classifiers using the predictability of those labels are known as pseudo-labelled data. Training and pseudo-labelling are the two distinct stages that make up the self-training process. The labelled data as well as potentially pseudo-labelled data from earlier iterations are used to train the classifiers during the training phase. The most reliable recommendations will be applied to the labelled data set for the following iteration after the classifiers are employed to determine labels for the unlabelled data points in the pseudo-labelling stage. Self-training can be applied and modified in many ways, including object identification (*Chen et al., 2022*) and hyperspectral picture classification (*Li et al., 2023*). Important design choices in self-training include choosing which data to pseudo-label, reusing pseudo-labelled data, and establishing ending conditions. Assuming accurate probabilistic predictions are provided, the procedure may resemble the expectation-maximization (EM) method. Non-probabilistic algorithms may require modifications, such as enhancing prediction probability estimations or through local distance-based metrics. In *de Vries & Thierens (2021)* the use of self-training in random forests is investigated and a termination criterion centred around out-of-bag error is provided. Furthermore, in *Chen et al. (2022)* a progressive method known as the pseudo-label technique is discussed in which data points are pseudo-labelled during the training phase. This method is specifically applicable to neural networks.

**Semi-supervised support vector machine (S3VM):** The volume of data collected globally across multiple technologies has increased dramatically in the past few years. Over this period, a plethora of technologies and methods for effectively gathering this information have been established. However, there are a lot of unlabelled data chunks in this acquisition, so labelling them all will take a lot of effort and time. In order to improve learning behaviour, the S3VM approach was explored in conjunction with support vector machines (SVMs), as semi-supervised learning techniques often focus on mixing labelled and unlabelled input (*Calma, Reitmaier & Sick, 2018*). ML techniques built around statistical learning theory are known as SVMs. They prove useful in several areas, including generalisation, simplicity of the solution, global optimisation, and an established theoretical basis. They are proven to be useful in real-world engineering domains. One constraint of SVM is that it is best used with supervised learning and, thus, only on labelled data. A learner model that can function well with little labelled data is quite uncommon. Because of this, semi-supervised learning may constitute a useful technique to use with

SVM in situations when labelled data is scarce but adequate unlabelled data exists (*Devgan, Malik & Sharma, 2020*). Finding the best classification hyperplane that meets classification objectives is the primary goal of SVM. An S3VM's concept is to construct a single, fundamental SVM for each labelling before selecting the SVM with the largest margin. These are created using the same methodology as SVM but with a few minor parameter adjustments and the addition of a penalty term to the objective function. Although SVM's aim is convex, the semi-supervised SVM's objective is non-convex and predominantly hat-shaped. Investigation of various S3VM optimisation methods may prove quite rewarding. Several well-known applications are Branch and Bound, SDP convex relations, deterministic annealing, SVM$^{light}$, ▽S3VM, continuation S3VM, and CCCP. S3VMs have several applications where they can be useful. These hold true anywhere that SVMs are utilised, thereby continuously enhancing SVM performance. Because it is based on a well-defined mathematical structure, it operates more quickly and requires less computing time. Since S3VMs utilise unlabelled data, they function more effectively than SVMs, which has made them increasingly prevalent in recent times. Even while they have been useful, there is a chance that they will lead to even worse performance than if you simply used labelled data. To prevent these types of issues, just those cases that are anticipated to be beneficial must be chosen, avoiding the cases that might harm the system's efficiency in general, rather than taking advantage of all the unlabelled data. These issues can be simplified using one of the two methods: S3VM-c or S3VM-p. Unlabelled data may prove useful when component density sets are discernible or observable, which is the only situation in which S3VM-c operates. This involves computing the label and confidence for every cluster in SVM and S3VM. Conversely, S3VM-p primarily relies on confidence estimates in graph approaches, such as those where confidence may be expressed as the threat involved in unlabelled cases. In each of these scenarios, the S3VM prediction findings are selected if they exhibit a high degree of confidence in the label propagation bias; if not, the SVM prediction is employed. Adopting S3VM-c and S3VM-p might lessen the likelihood of performance loss nevertheless are certain drawbacks to its use.

**Generative adversarial networks:** Generative adversarial networks (GANs) are a new kind of learning framework that was most recently presented. They operate on the principle of concurrently building generative and discriminative learners (*Goodfellow et al., 2020*). This method, which is typically applied with neural networks, trains a discriminative classifier to determine if a given data point is "real" or "fake" (*i.e.*, artificially generated) at the same time as a generative model, which is tasked with producing data points that are challenging to differentiate from actual data. Two neural networks, a discriminator, and a generator, participate in a competitive training procedure to form GANs. The generator creates synthetic data using random noise as input with the goal of producing samples that are identical to actual data. The discriminator continuously adapts to better separate real from synthetic data while it is being trained to do so concurrently. The discriminator gains proficiency in identifying authenticity, while the generator improves its capacity to produce realistic data through recurrent adversarial training. Stability is reached as an outcome of the above dynamic process, where the discriminator has a 50% probability of accurately categorising the outstanding synthetic data that the

generator provides. Using a game-theoretic methodology, GANs enhance efficiency across the discriminator and generator through adversaries (*Sreevallabh Chivukula et al., 2022*). Their learning is guided by their desired function; the discriminator's goal is to correctly categorise synthetic and real data, while the generator aims to maximise the possibility of deceiving the discriminator. After being trained, GANs may produce new, genuine samples that closely resemble the training set. This makes them useful for a variety of tasks, including data augmentation, picture production, and style transfer.

## Ensemble models

**Support vector machine:** Regression and classification challenges are handled using this ML model. Support vector regression is a variant of SVM's fundamentals to predict continuous values. SVR aims to minimize the margin of error and determine the optimum hyperplane to match the data. The SVR algorithm is designed to minimise both the level of sophistication of the model and the empirical risk, which is defined as the difference between the values that are predicted and those that occur. To accomplish this, a margin of tolerance ($\varepsilon$) is introduced, which penalizes data points that exceed a particular error margin while permitting certain data points to remain within it. To enable SVR to process non-linear connections, the kernel technique is frequently utilised to translate data into a higher-dimensional feature space (*Berk & Berk, 2020*). Polynomial kernels and the Radial Basis Function (RBF) are examples of popular kernel functions. The efficiency of SVR is influenced by the kernel function selection and its hyperparameters. SVR's interpretability, scalability, and ability to handle big datasets have all been the subject of recent studies. An optimization issue is addressed, usually using gradient-based approaches or sequential minimal optimization (SMO) techniques, to train an SVR model. SVR is used by *Lee et al. (2020)* to anticipate dependability in engine components, where he suggests an innovative method that uses legitimate genetic algorithms to find SVR's best parameters, and then uses those values to build SVR models. Non-linear regression and time series issues have been successfully resolved using SVMs (*Sexton et al., 2017*; *Chang & Lin, 2011*). In contrast to conventional neural network models, which employ the empirical risk minimization principle, SVMs employ the structural risk minimization principle, which aims to reduce an upper bound on the generalisation error rather than the training error. According to this theory, SVMs are more generalizable than conventional neural networks. In various disciplines, SVMs have been used to solve predicting issues. SVMs have, however, seldom been used to predict reliability. *Malhotra et al. (2021)*'s study represents the preliminary approach to using an SVM model to forecast engine reliability, with the goal of assessing the viability of SVMs in reliability predicting. SVMs are used by *Chou & Truong (2019)* to evaluate software dependability with simulated annealing algorithms (SVMSA), which yields more accurate forecasts compared to the different techniques. In comparison to ANN-based and current SVM-based models, *Chou & Truong (2019)*'s suggested SVM-based software reliability prediction model may attain a stronger prediction precision.

**Decision tree:** A non-linear supervised learning approach called decision tree regression is utilised for both regression and classification problems. Decision trees are used in regression to predict the target variable by iteratively dividing the dataset into

subsets and making decisions at all nodes. This is done depending on the input attributes. Averaging all the target values at the leaf nodes is the eventual estimation (in the case of regression). Decision trees render insight on the significance of features, which aids in locating important elements that lead to breakdowns in equipment. Its interpretability is essential for comprehending the process of making decisions. Based on the collected historical data of the equipment's operation and performance from sensor readings, maintenance logs, and any other appropriate data, the model provides inferences on the machine's condition and degradation. The data may show anomalous patterns that point to malfunctions or upcoming breakdowns. The model can innately identify the non-linear relationships and categories them as complex degradation patterns, thus indicating possible problems with equipment or machinery. Mean absolute error (MAE) or root square error (RMSE) metrics can be used to estimate how accurate the prediction of the model is.

**Random forest:** It is a ML algorithm designed by *Breiman (2001)* designed RF, which can be used for regression and classification tasks. It is an ensemble learning method that constructs a multitude of decision trees at training time and outputs the class that is the mode of the classes (classification) or mean prediction (regression) of the individual trees. Reducing overfitting and increasing prediction accuracy are two advantages of this ensemble technique. Random subsets of the data and features are used to train each decision tree in the ensemble. Because of this unpredictability, the model is more resilient to noise and data outliers, and the correlation between the trees is decreased. The average of all the trees in the ensemble's forecasts yields the final forecast. Every RF decision tree is trained using a bootstrapped subset of the initial training data, which implies that each tree sees marginally distinct data points, in order to minimize overfitting. Furthermore, RF only takes into account a random subset of characteristics at every split when dividing nodes in decision trees. This decreases the possibility of overfitting and further extends the trees. The generalisation inaccuracy for forests converges as the number of trees in the forest rises. The RF has significant additional advantages. Predicting the state of a robust automated facility uses ML techniques like linear regression, RF, and symbolic regression (SR) (*Zenisek, Holzinger & Affenzeller, 2019*). They put forth a strategy for identifying and forecasting idea drifts—drifting behaviour—in uninterrupted data feeds. A scenario-based analysis on manufacturing radial fans was also given. Approaches, drift detection, and prediction showed accurate outcomes, according to the outcomes produced using the simulated data. Additionally, researchers located at the strained radial fan revealed that the congestion detection technique was successfully used based on the actual research that was carried out. Although the analysis results were encouraging, the forecasting of notion variations presently depends on the assumptions and cannot be assessed due to the absence of ongoing degradation data.

For continuous conditioning and problem detection in rotating machinery, *Janssens, Loccufier & Van Hoecke (2019)* developed a multi-sensor system that combines vibration measures in addition to infrared thermal imaging data. Prototype characteristics are created using vibration measurements, while statistical characteristics are generated using infrared thermal imaging data, using a technique called feature synthesis. For practical

defect identification, the collected characteristics are then pooled and given to RF classifiers. Researchers have shown in the investigation that combining both types of sensor information paves the way for more precise measurement of a wide range of cases and defects than when using a single sensor feed. A learning method that can accurately identify and analyse heterogeneous datasets of turbofan engines was created by *Lacaille & Rabenoro (2018)*. The proposed model employs a vast proportion of pre-trained and statistical tests on the dataset, and it is capable of choosing effective sets of tests having pre-identification rates of more than 85%. A new technique to use RF to identify damaged rotor bar breakdown in a Line Start-Permanent Magnet Synchronous Motor (LS-PMSM) was put forth by *Quiroz et al. (2018)*. A working motor and a damaged motor were used to collect the instantaneous current signal while setting up the motor. Thirteen distinct analytical time-domain features were employed to extract features for the model's training, and later all these features were utilised to identify the status of the motor when it is running properly or improperly. To choose only a few features from the RF, they took into account the relevance of each function. According to the findings, RF accurately classifies motor disorders as healthy or inadequate with a 98.8% confidence when using all characteristics and a 98.4% accuracy when employing only the mean index and impulsion features. The generated framework was compared to various well-known ML techniques, such as decision tree, naive Bayes classifier (NBC), linear ridge, and SVM. The RF regularly surpasses these methods, outperforming them with greater precision. The recommended approach can be employed in the industry for continuous monitoring and fault diagnosis of LS-PMSM machines, and the results may be useful for creating preventative maintenance schedules in manufacturing facilities.

By examining the historical information of aviation safety systems, *Yan & Zhou (2017)* postulated a prediction model using Term Frequency-Inverse Document Frequency (TF-IDF). He also stated that RF can estimate failures of great specificity well ahead of time, and corrective actions could be conducted based on the model's predictive accuracy. The previous aircraft in a row have used TF-IDF to generate the attributes from the unprocessed data. The developed RF model's classification of the problems took into account various priorities. As a result of the dataset's extreme disparity, the receiver operating characteristic (ROC) curve was selected as a benchmarking tool. The proposed methodology achieves the highest true positive value of 100% and the least false positive rate of 0.13% when matched to some of the other approaches. The suggested technique obtains a true positive rate of 66.67% and a false positive rate of 0.13% for the test sample.

**Gradient boosting machine:** This model is one of the ensemble ML techniques for regression and classification problems. It builds a series of weak learners sequentially, which are basically decision trees that have a small number of nodes and are usually shallow. At every stage, the model aims to rectify the error or the residuals of the previous one. Every data I assigned with a weight in this model. To maximize the overall efficiency of the approach, gradient boosting regression fits a new model to the errors of the earlier model. The learning rate is one of the hyperparameters that manage the impact of each weak learner until the final prediction. The model is more resilient when the learning rate is lower, nevertheless, it needs more trees for a convergent result. The weighted aggregate

of all the estimates made by the weak learners is the ultimate prediction (*Aziz et al., 2020*). The metrics used for the Decision Tree can be considered for evaluating the efficiency of the Gradient Boosting model.

**Light gradient boosting machine:** An ensemble ML technique called Light GBM serves to address challenges with regression as well as classifications. Despite this, a boosting family including GBM and Adaboost is nevertheless employed since it is important to generate effective classifiers from the weak ones. Several researchers have begun to build and advance toward Light GBM and its variants due to the obvious benefits of focusing on data weighting (*Aziz et al., 2020*). Researchers have used Light GBM extensively to attain state-of-the-art outcomes on a variety of machine-learning studies, including predicting the types of breakdowns of reinforced concrete panels (*Hoon et al., 2018*), identifying malfunctions in the robot sectors (*Liang et al., 2019*; *Tang et al., 2020*), and accurately classifying airline data (*Ye et al., 2019*), home credit datasets, and financial sectors (*Aziz et al., 2020*). The majority of the researchers use learning rate, number of leaves, depth, and tree count to create the majority of the hyperparameters for Light GBM. The accurate tailoring of hyperparameters to reduce overfitting is crucial for Light GBM's accuracy in predictions. In contrast to the GBM family of boosting algorithms, which includes Adaboost, Light GBM possesses unique advantages. It offers additional benefits, such as distributed learning, improved training efficiency, and support for parallel learning to handle massive data sets (*Aziz et al., 2020*). Both GBM and Light GBM are ML techniques that are frequently employed in the field of prediction. For example, they are used to identify the seismic failure mode of concrete shear walls (*Song et al., 2019*) and to forecast bubble points (*Dev & Eden, 2019*) in order to get valuable insights specific to the field. *Li et al. (2018)* presents an innovative method that uses the Light Gradient Boosting Machine (LightGBM) algorithm to estimate the RUL of aviation engines. Using actual aircraft engine data, the study reviews the suggested method and finds that it performs better than current RUL estimate techniques in terms of accuracy and computing efficiency.

**Extreme gradient boost:** This model is categorized among boosting techniques, which aim to enhance the efficiency of the model by creating strong classifiers from weak classifier combinations (*Aziz et al., 2020*; *de Oliveira, 2019*). Benefiting from an effective and refined version of the three boosting techniques, Adaboost, GBM, and Light GBM, XGBoost produced excellent outcomes across multiple benchmark dataset classifications, including the credit scoring domain (*Daoud, 2019*), biochemical index prediction (*de Oliveira, 2019*), and genome segment-based breast cancer prediction (*Xia et al., 2017*). The basic principle of this approach is to expand the size of decision trees by optimizing them by applying gradient descent when new decision trees are included for training. For regression problems, predictions are combined with the weighted sum, and for classification, with the majority voting. All the algorithms covered in this study under the ML algorithms have been represented in the form of a chart in Fig. 4.

## Condition monitoring techniques in PdM

The primary uses of AI in maintenance schedules are fault diagnostics and anomaly identification. The cause of a failure can be determined through fault diagnostics. Only the

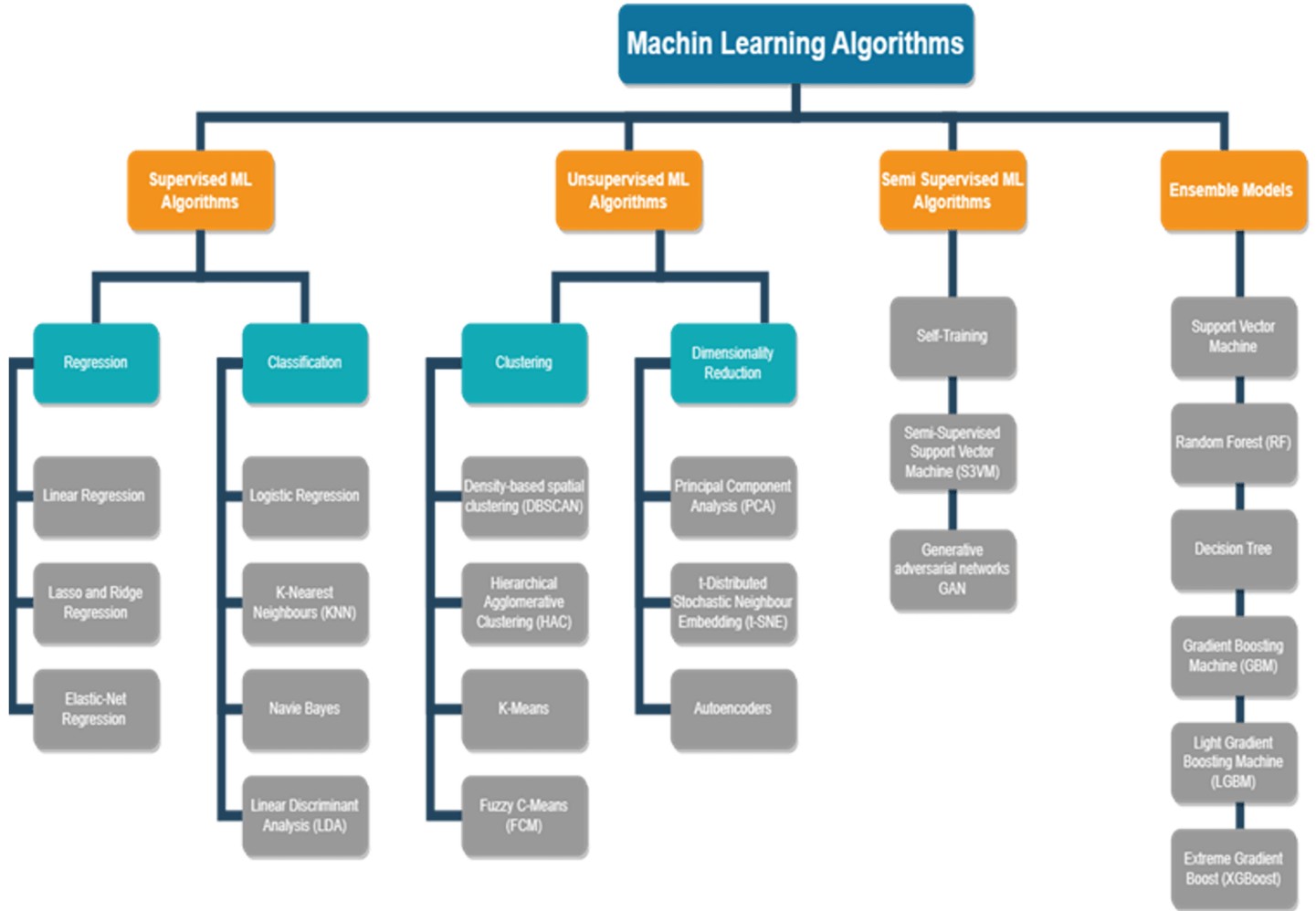

**Figure 4 The machine learning algorithms.** A comprehensive overview and a visual representation of various machine learning algorithms categorized by their learning paradigms such as supervised, unsupervised, and semi-supervised learning.

frequency of a failure can be determined by anomaly detection, but it provides benefits for dataset production. The above viewpoints are used in this segment to examine the literature.

### Fault diagnostics

Equipment condition maintenance is constantly concerned with determining the connection between monitoring data and equipment health status, and fault detection is crucial in resolving such problems (*Gao, Cecati & Ding, 2015*). The inspection and repair of devices and industrial plants has made extensive use of ML to carry out fault diagnostics (*Lei et al., 2020*).

A brand-new out-of-distribution (OOD) detection-assisted reliable equipment failure diagnostic approach was devised by *Han & Li (2022)*. A deep ensembled fault diagnosis system is initially built by fusing various deep neural networks. Then, to identify OOD data

and provide alerts on possibly suspect findings, a dependability analysis is carried out using a confusion depth ensemble. To make reliable conclusions, the prognosis and ambiguity of the deep ensembles are rigorously taken into account. A gearbox condition monitoring example and a wind turbine predictive control instance were used to evaluate the suggested approach. The sample data for wind turbine defects comprises three groups of defects and one group of normal data, each with 1,000 samples. The outcome shows that it offers notable benefits in identifying OOD samples and achieving reliable malfunction detection. For the identification of rolling bearing faults, *Liu et al. (2022)* presented a deep feature-enhanced deep convolutional network (GAN). To prevent mode breakdown and boost the reliability of the GAN, a novel generator objective factor coupled with a pull-away function was created. The GNN makes utilise a self-attention approach to speed up the understanding of the characteristics of the underlying vibration signal. An automated dataset extractor was built to ensure the precision and heterogeneity of the sampling that was produced. The final component is the addition of a convolutional neural network as a classifier for defect diagnostics. A rolling bearing vibration signal dataset from an electrical locomotive was used to test the approach. The sample consists of five types of failure statistics with a study population of 12,600 and one group of normal data with a study population of 126,000. The outcomes show that this strategy performs best in imbalanced sample fault detection and diagnosis. A failure detection approach for rotating shafts based on the Wasserstein distance was proposed by *Ferracuti et al. (2022)*. The Wasserstein distance was taken into account in the training phase to distinguish between the various equipment operating circumstances by the authors after they retrieved frequency- and time-based features from the vibration signals. Without building a classifier, the empirical distance-based fault detection methodology allows the determination of the failure characteristic. As a result, it is highly effective and suitable for embedded hardware. Under a variety of operational situations and poor signal-to-noise ratios, this approach can resolve the issue of defect diagnostics for rotating machinery. It is also capable of system prediction and diagnostics, enabling rotating machinery preventative analysis. The issue of induction motor malfunction identification and diagnosis using the motor's current pattern analysis was investigated by *Ferracuti et al. (2015)*. Using a Clarke-Concordia transformation and kernel computation, the studies found the cumulative distribution functions of data pertaining to both functional and defective motors. Since Kullback-Leibler divergence shows the difference in two probability distributions, it is employed as an indicator for the detection and recognition of vulnerabilities. Kernel density prediction is improved *via* fast Gaussian transform. This approach allows for real-time quality assurance by the end of the manufacturing line and has a minimal computing overhead. According to the results of the studies, the suggested approach is capable of identifying and diagnosing a variety of induction motor problems and malfunctions.

As observed from the aforementioned evaluations, ML models must be trained on faulty datasets in order for fault detection approaches to work. Based on the collected data, the predictive model can detect the existence of a failure and classify the kind of defect. Nevertheless, there are restrictions on the applicability of this technology because it is challenging to gather enough failure data in a genuine test environment.

### Anomaly identification

One of the major concerns in the manufacturing industry is the poor rate of pre-emptive anomaly identification of machinery (*Gama, Ribeiro & Veloso, 2022*), leading to undetected and overlooked anomalies. Effective anomaly identification techniques detect variables that show deviation from the regular samples in the dataset. Identifying such anomalies at the initial stages will reduce the equipment breakdown considerably. Enormous data produced by the manufacturing industry, require a monitoring system that constantly watches for patterns that deviate from normal behaviour thereby giving prompt alarms for anomalies.

In *Yan et al. (2023)* deep transfer learning for time series anomaly detection has been categorised into three different approaches such as parameter transfer, instance transfer, and adversarial transfer. Among these approaches, for industrial applications for time series anomaly identification, parameter transfer is the highly preferred method. In *Choi et al. (2022)*, self-supervised learning model, unsupervised/semi-supervised learning models, multivariate time series data, and temporal dependencies are given great emphasis. The researchers propose feature selection, transformation of time series data, and the use of deep learning models for handling time series data. In *Singh & Singh (2020)*, feature vectors transmitted by sensors are proposed as a methodology for overcoming the challenges in implementing IoT solutions in a resource-constrained industrial process. The trade-off between distortion and the ability to discriminate normal and anomaly data has been explored.

In order to identify bearing failures, *Zhao et al. (2017)* suggested a single-class classification algorithm utilising the extreme learning machine boundary (ELM-B). Including an input layer, a hidden layer, and an output layer, the prototype is a single-layer feed-forward neural network. The RMS, kurtosis, peak-to-peak, crest factor, and skewness of the good bearing vibration signals were evaluated and used as input variables. The outcome of the system is developed to be a single 1. Sensing devices gather and provide the bearings' vibration signals to the learned model. It is considered that a failure has resulted if the outcome does not equate to 1. An approach for identifying irregularities in conveyor carrier wheel bearings in auto manufacturing lines was put out by *Tanuska et al. (2021)*. Only 18 anomalies were found in the 16,000-bearing temperature information that the investigators obtained. With 13 neurons in the input layer, 18 neurons in the hidden layer, and two neurons in the output layer, the authors created a multi-layer perceptron (MLP). Depending on the bearing's optimum and mean temperature, the MLP can identify irregularities in the bearing. Convolutional autoencoder (CAE)-based anomaly detection methodology was provided by *Kähler, Schmedemann & Schüppstuhl (2022)* to identify superficial flaws in aviation systems. The landing gear surfaces of aircraft were photographed by the authors in 600 fault-free photos and 300 faulty images. An additional 100 fault-free images and the 300 faulty images are employed for evaluation, while 500 of the fault-free images would be randomly chosen from this group for learning.

The work mentioned here leads to the conclusion that outlier detection methodologies can help with the industry's issue of collecting very few defect samples since the technique

needs regular training data. Anomaly detection's drawback, meanwhile, is that it does not pinpoint the precise reason for a problem. The following table, Table 2 gives a brief overview of the various algorithms and models used in I4.0 for PdM.

### Explainability and interoperability in PdM

Explainability pertains to the methodologies and approaches employed within the domain of ML to furnish coherent and significant reasoning regarding how a model generates predictions or decisions. These explanations function as a conduit connecting the intricate internal mechanisms of the model and human users, facilitating their understanding of the logic that underlies the model's outputs. Conversely, interpretability within the domain of ML concerns the degree to which a layperson can comprehend and place confidence in the model's prognostications or determinations. An interpretable model is characterised by its internal mechanisms, including parameters and decision processes, being readily comprehensible and deducible by humans. This allows them to acquire insights into the model's operation and the rationale behind its generation of specific outputs.

*Cummins et al. (2024)* examines several explainable techniques employed in the domain of PdM. The techniques may be classified into two distinct categories: explainable methods that are particular to the model and explainable methods that are not dependent on the model. The generation of explanations by model-specific approaches, such as CAM, GradCAM, and DIFFI, is predicated on the qualities of the architecture they want to elucidate (*Kumar, Zindani & Davim, 2019*). Conversely, model-agnostic techniques, such as SHAP and LIME, provide the capability to be employed across many architectures and offer explanations that are not contingent upon the particular model. The publication furthermore presents a table that offers a summary of many explainable approaches found in the literature. These methods include Shapley Values, LIME, Feature Importance, LRP, Rule-based, CAM and GradCAM, Surrogate, Visualisation, DIFFI, Integrated Gradients, Causal Inference, ACME, and Statistics. The belief that the predictive performance of a predictor is inversely proportional to its degree of interpretability is contested in *Leblanc & Germain (2023)*. The article presents an analysis of the Rashomon set, an assemblage of predictors with roughly equivalent accuracy, and posits that explainability and interpretability are complimentary instead of substitutive.

The assertion that the predictive performance of a model is directly proportional to its complexity (Belief 2) is called into question and challenged as the phenomenon of double descent in deep learning and its potential corroboration with Belief 2 is investigated. By utilising explainable AI techniques to monitor the loss of the autoencoder in relation to the input data, anomalous signals is identified in *Serradilla et al. (2021)*. In order to ascertain the underlying cause of anomalies identified in the training data, XAI-based diagnosis tools are created to analyse them. Enabling experts to recognise anomalies in a semi-supervised fashion, XAI provides a diagnostic and interpretative instrument for aberrant data points. By increasing the interpretability and transparency of data-driven models, XAI facilitates the implementation of data-driven PdM systems in smart factories. Descriptions for the identification of anomalies and the diagnosis of malfunctions in press machine data are furnished by XAI techniques.

**Table 2 Recent applications of ML algorithms in predictive maintenance.** The various other ML techniques used in predictive maintenance are mentioned with the key findings.

| ML techniques | Equipment/system | PdM data | Key findings |
|---|---|---|---|
| ANN (*Hesser & Markert, 2019*) | Tool wear of a milling machine | Acceleration data | -The ANN, SVM, and KNN classifier's score, recall, and precision are computed.<br>-Utilizing ANN, it is possible to calculate the device's RUL.<br>-Extended acceleration data will cause the instrument to degrade gradually. |
| ANN, SVM (*Falamarzi et al., 2019*) | Rail-tram track | Data-gauge measurements data | -In terms of forecasting the gauge deviation from straight segments, ANN model outperforms SVM, whereas SVM provides superior forecast for curved segments. |
| Logistic regression (LR), Extreme gradient boosting (XGBoost), RF (*Binding, Dykeman & Pang, 2019*) | Printing machine | Machine's status data | -In regard to evaluation criteria, RF and XGBoost outperform LR.<br>-With regards to ROC, all the techniques fared comparably higher. |
| RF, DT, NBC, SVM, LR, linear ridge (*Quiroz et al., 2018*) | Rotor bar | Transient current signal data | -In the LS-PMSM, RF is employed to identify damaged rotor bar malfunction.<br>-Outcomes from RF classifiers fared better than those from other approaches.<br>-The RF method's accuracy and dependability were confirmed by its 98.8% accuracy in defect analysis. |
| Supervised aggregative feature extraction (SAFE) (*Yan & Zhou, 2017*) | Ion implementation tool | Maintenance cycles dataset | -Time-series feature extraction for PdM was handled by the SAFE approach, it fared better than the conventional feature extraction techniques. |
| SVM-PF, genetic algorithm particle filter (GAPF) (*Guo & Sui, 2019*) | Aircraft | Actuator internal oil leakage fault data | -SVM-PF has greater predictive correctness and is a new adaptive methodology for oil leakage malfunction prediction for actuators. –In comparison to conventional ML, a better degree of error correction was seen. |
| Dynamic regression (DR) (*Ahmad et al., 2019*) | Rolling element bearing | Vibration measurement data | -Regression models were used to predict the health and RUL of the rolling element bearing.<br>-Used alarm bound technique (ABT) to determine time to start prediction (TSP).<br>-Achieved a reasonable result compared to other techniques. |
| DT, ANN, RF, gaussian naïve bayes (GNB), bernoulli naïve bayes (BNB) (*Kolokas et al., 2018*) | Anode production of industrial equipment | Process sensor data | -PdM approach for real-time defect prediction even before a breakdown occurs.<br>-Device problems are predicted 5–10 min in advance. |
| Multi-gene genetic programming (MGGP) (*Garg et al., 2015*) | Metal lathe machine | Vibration and acoustic signals | -A unique sophistication metric-based MGGP methodology for failure analysis is presented.<br>-The system works better than traditional MGGP variants. |
| Hierarchical clustering (HC), K- medoids, K-means (*Yoo, Park & Baek, 2019*) | Chemical vapor deposition (CVD) process | CVD sensor data | -Incorporating sensor information from the CVD process, predictions were verified.<br>-Ordering Points To Identify the Clustering Structure (OPTICS) showed improvement compared to the other methods and can therefore be employed for CBM decision-making. |
| Autoregressive integrated moving average ARIMA, generalized born (GB), RF and recurrent neural network (RNN) (*Jimenez-Cortadi et al., 2020*) | Spindle load | Piece number and tool position | -Created for RUL modelling using a linear regression approach.<br>-Increasingly accurate results were collected in order to forecast the RUL for comparability. |

| ML techniques | Equipment/system | PdM data | Key findings |
|---|---|---|---|
| Gradient boosting machine (GBM), K-nearest neighbors, BNB (*Manfre, 2020*) | Engine equipped with a rotating shaft | Vibration, noise, pressure, temperature, humidity | -The isolation forest algorithm delivered the best findings when the learning duration was 20% longer. |
| LR, decision tree regression (DTR), random forest regression (RFR), SVR (*Luo et al., 2020*) | Machine tool | Vibration, force, temperature, federate, cutting depth | -With a slight inaccuracy, the hybrid technique powered by the digital twin estimates results that are rather identical to the true value and offers more exact projected results, 6.27% at the late stages. |
| NBC, deep neural network (DNN), convolutional neural network (CNN), deep transfer learning (DTL) (*Janssens et al., 2018*) | Rotating machinery | Accelerometer, thermocouple, and thermal camera measurements | -The CNN algorithm was used to enhance online CM in offshore wind turbines and identify different rotating equipment faults.<br>-Can be utilized to inspect bearings in production processes.<br>-CNN surpasses in both device faults diagnosis and oil condition estimation.<br>-The Feature learning (FL) method adds 6.67% more than the Finite element (FE) method. |
| RF, TF-IDF -SVM, LR (*Quiroz et al., 2018*) | Aircrafts | Historic data of aircraft maintenance system | -TF-IDF was employed to derive the attributes from the raw flight data.<br>-With a minimum of 0.13%, RF obtains a true positive rate of 66.67%. |
| Long short-term memory (LSTM) (*Aydin & Guldamlasioglu, 2017*) | CMAPSS NASA engine | Simulation dataset of engine degradation | -Manages time series dataset, and the outcome of a LSTM model is used to assess a device's condition prior to the expiration of its lifespan and assist in anticipating failures.<br>-Trained and validated using free source engine deterioration data. |

XAI techniques are incorporated to aid specialists in failure diagnosis by identifying signal components that are accountable for anomalies and supplementing their knowledge in this regard. In summary, the incorporation of XAI methods into PdM approaches indicates a significant progression within the domains of asset management and industrial maintenance. By augmenting the visibility and comprehensibility of ML models implemented in PdM systems, XAI facilitates practical understanding of the fundamental causes that contribute to equipment deterioration or failure.

## CONCLUSION

The literature review is categorized based on the current maintenance technique adopted by the manufacturing industries after the three industrial revolutions. The literature mainly draws attention to the role of ML in PdM in order to reduce the downtime of the industry thereby increasing its production rate. The literature gives an in-depth understanding of the PdM planning model, where data cleaning, data normalisation, optimal feature selection, decision models, ML algorithms, and predictive models have been explained in detail. According to a thorough study of the research, PdM is still a crucial strategy for boosting productivity in any setting in which there are devices that degrade with time. With the development of IoT, the potential for producing and installing

inexpensive, linked sensors will grow. ML techniques may be used to do PdM as the quantity of information and the number of sensors rise. A thorough analysis of ML approaches used in the PdM of manufacturing equipment is presented in this work. It has been noted that ML is an inventive way to apply PdM and that there are huge business prospects for it. Only 11% of the organisations surveyed in a PwC report, however, claimed to have "realised" PdM based on ML (*Seebo, 2019*).

The top popular ML algorithms in the studied work are SVM, RF, and ANN. They have been used effectively in some PdM application fields. On ANN ML algorithms, certain writers (*Cheng et al., 2020*; *Hesser & Markert, 2019*; *Falamarzi et al., 2019*; *Binding, Dykeman & Pang, 2019*; *Janssens et al., 2018*; *Huuhtanen & Jung, 2018*; *Kumar, Shankar & Thakur, 2018*; *Koca, Kaymakci & Mercimek, 2020*; *Aremu, 2019*; *Fernandes et al., 2020*; *Calabrese et al., 2020*; *Hoffmann et al., 2020*) concentrated their work. Other writers (*Janssens, Loccufier & Van Hoecke, 2019*; *Zenisek, Holzinger & Affenzeller, 2019*; *Janssens, Loccufier & Van Hoecke, 2019*; *Lacaille & Rabenoro, 2018*; *Quiroz et al., 2018*; *Yan & Zhou, 2017*; *Aziz et al., 2020*; *Hoon et al., 2018*; *Carbery, Woods & Marshall, 2019*; *Butte, Prashanth & Patil, 2018*; *Amihai et al., 2018*; *Paolanti et al., 2018*; *Kulkarni et al., 2018*; *Su & Huang, 2018*; *Dos Santos et al., 2017*; *Keartland & Van Zyl, 2020*) researched RF methods. Authors have recently focused on the use of the SVM technique (*Luo et al., 2020*; *Kaparthi & Bumblauskas, 2020*; *Xiang, Huang & Li, 2018*; *Du et al., 2018*; *Lasisi & Attoh-Okine, 2018*; *Mathew, Luo & Pang, 2017*; *Arias, 2017*; *Ha et al., 2018*; *Eriksson, 2020*). The GBM technique has received attention from certain authors (*Calabrese et al., 2020*; *Butte, Prashanth & Patil, 2018*; *Xiang, Huang & Li, 2018*; *Eriksson, 2020*). Some authors (*Lacaille & Rabenoro, 2018*; *Quiroz et al., 2018*; *Luo et al., 2020*; *Kaparthi & Bumblauskas, 2020*) used the DT approach to conduct investigations. However, it has been noted that the XGBoost technique is not given as much weight and that there is not much research using it in the literature (*Calabrese et al., 2020*; *Carbery, Woods & Marshall, 2019*; *Sreevallabh Chivukula et al., 2022*). According to the literature, the RF algorithm is the most widely used ML technique for PdM because it has been used on a wide range of industrial devices, systems, or components, such as rotating machinery, turbofan engines, rotor bars-LS-PMSM, airliners, manufacturing units, integrated circuits, industrial pumping systems, cutting machines, grocery store conditioning systems, hard drive discs (HDD), wind turbines, vending machines, and numerous others. The most popular ML approaches, commercial implementations, and data types for ML applications are all identified in this study, which also makes recommendations for the future and lays the groundwork for future investigations. Future work in the area of PdM under I4.0 will require an integrated strategy that takes into account numerous elements of ML and planning models. To begin, advances in the merging of ML and deep learning approaches should be investigated in order to fully realise the possibilities of complex and nonlinear interactions in industrial data. Exploring innovative systems and techniques capable of effectively handling the multidimensional and time-series characteristics of PdM data will be critical to producing more accurate and reliable forecasts.

Furthermore, there exists an imperative to explore further FDD systems, with an emphasis on creating hybrid models that combine data-driven techniques with expertise to

improve the understanding and dependability of problem-detecting methods. A more in-depth investigation of the incorporation of various explainable AI approaches into FDD systems may lead to further transparent and practical findings for maintenance professionals.

In addition, future research should focus on evaluating and assessing PdM models. The establishment of standardised criteria and standards for analysing the efficacy of numerous techniques in various industrial contexts would help to create an increasingly uniform and comparative environment in the sector. Due to the dynamic character of manufacturing processes, continuous degradation monitoring and operational impact modelling should be investigated. This entails creating models that can adapt and grow over time to reflect changes in operating conditions and system behaviour.

We can consider the possible advantages of edge-cloud integration for maintenance prediction. Determining how to use edge computing devices for early data preparation and real-time decision-making can eliminate the requirement to send vast amounts of unprocessed information to the cloud. This method may improve the overall effectiveness of the PdM system.

### Funding
The authors received no funding for this work.

### Competing Interests
The authors declare that they have no competing interests.

### Author Contributions
- Ida Hector conceived and designed the experiments, performed the experiments, analyzed the data, performed the computation work, prepared figures and/or tables, authored or reviewed drafts of the article, and approved the final draft.
- Rukmani Panjanathan conceived and designed the experiments, performed the experiments, analyzed the data, performed the computation work, prepared figures and/or tables, authored or reviewed drafts of the article, and approved the final draft.

### Data Availability
This is a literature review.

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
