# Peer review of "Predictive maintenance in Industry 4.0: a survey of planning models and machine learning techniques"

_PeerJ Computer Science, doi:10.7717/peerj-cs.2016_

## Round 0.1 · original submission · Major Revisions

Thank you for submitting your manuscript titled “Predictive maintenance in industry 4.0: A survey of planning models and machine learning techniques” to PeerJ Computer Science. I have now carefully reviewed the feedback from the two reviewers. Both reviewers provided detailed comments which emphasize the significance of revision in different aspects of your manuscript.
Major Points:
1. Structure and Content: Both reviewers expressed concerns about the organization, structure, and content of the paper. In particular:
• Reviewer 1 mentioned taxonomy-related issues, repetition, definitions, and inconsistencies.
• Reviewer 2 recommended changes to the article's structure to better align with branches of Machine Learning and highlighted concerns about content overlap with an earlier review by Carvalho et al. 2019.
2. Clarity and Precision: Several remarks have been made regarding the clarity and precision of the definitions and terminologies used, especially concerning machine learning and predictive maintenance. Both reviewers pointed out issues with the definition of ML, with Reviewer 2 suggesting reference materials to refine it.
3. Survey Methodology: Reviewer 2 expressed concerns about the study's replicability due to insufficient details in the survey methodology. The choice of databases, keywords, and the justification for certain methodologies were found lacking.
4. Grammar and Language: Both reviewers found various instances where the language needs refinement. This includes subject-verb mismatches, repetitive definitions, and general grammar and formatting issues.
5. Referencing and Citation: Both reviewers found areas where additional citations were necessary to back statements or where clarification was required.
Recommendation:
Considering the extent of the revisions suggested by the reviewers, I recommend a major revision for your manuscript. I encourage you to carefully address all the points raised by the reviewers and to ensure that each point is systematically addressed in your revised version.
Once you have revised the manuscript, please provide a detailed response letter indicating how each comment was addressed. This will greatly help in the re-evaluation process.
We understand that a major revision requires substantial effort, but we believe that addressing these concerns will significantly enhance the quality and impact of your work.
Thank you for considering PeerJ Computer Science for your work. We look forward to receiving your revised manuscript.
Warm regards,

José Manuel Galán

**Language Note:** The Academic Editor has identified that the English language must be improved. PeerJ can provide language editing services - please contact us at copyediting@peerj.com for pricing (be sure to provide your manuscript number and title). Alternatively, you should make your own arrangements to improve the language quality and provide details in your response letter. – PeerJ Staff

Reviewer 1 ·

Basic reporting

1. Line 21: “, 4.0 Industry (I4.0)” to Industry 4.0
2. The authors need to rethink this statement in line 55-56 “The statistics gathered can also be used to create more effective approaches for intelligent PdM operations, often known as preventative maintenance.” Based on authors statement in Section 1.2, line 70, Preventive maintenance is an older technology and less intelligent than PdM. Thus this statement is not reasonable.
3. In line 94-97, the definition of machine learning is too vague and cannot represent the current ML technologies. The authors should consider rephrasing it.
4. In line 151 to 153, “PdM, also known as condition-based maintenance, aims to find patterns in component…”, this definition of PdM is controversial to the taxonomy of maintenance techniques stated by the authors in Section 1.3 and 1.4. In addition, as the definition and significance of PdM is given in section 1.4, it’s unnecessary to repeat or give a different definition of PdM or repeat the importance of PdM in many other places in the text, such as line 311, line 316 to 318, line 369 to 373.
5. Line 157 to 158: “Two techniques are employed to gather condition…”, the authors did not specify the two techniques.
6. Line 159, ‘registers’ seems not a proper wording here, consider ‘data logs.’
7. Line 161: What is “FMM operation”?
8. Line 167, ‘identical datasets…’, what are identical datasets and why are they removed in normalization?
9. Line 168 to 170, “Because prediction values can fluctuate across given bounds, machine learning algorithms must be complicated in order to produce accurate result.” Any reference to this statement? What’s the reason that fluctuations of prediction values require ML to be complicated?
10. Line 176, “outliers are typically seen as imbalanced datasets”, this definition of outlier is not right, and the authors need to reconsider it.
11. Line 177, “a number of outlier identification models have already been presented.”, It is suggested that authors can add some references.
12. The authors need to unify the usage of ‘normalization’ or ‘normalisation’.
13. Line 203 to 205, “Although the optimal feature selection decreases the number of incorrectly chosen attributes by half, it makes sure to 205 maintain the degree of true positive value.” Any reference to support this statement?
14. Line 208, ‘feature’ to ‘Feature’.
15. Line 245 to 247, the definition of different supervision level of ML is not accurate.
16. Line 309, two [33]
17. Line 313, there are two viscosity.
18. Line 322, four interconnected stages, but only three are given.
19. Line 370 to 373 is the similar to Line 311 to 313. This is redundant.
20. Line 383, ‘situation’ and ‘circumstances’ are the same meaning, only one should be kept.
21. Redundant definition of abbreviations in Line 378 to 380, Line 422 to 423, Line 502, and many other places. The authors should proofread the paper carefully. If an abbreviation is defined, there is no need to repeat it many times throughout the paper.
22. Line 474, capitalize the first letter.
23. Line 477, there are two periods.
24. Line 576 to 580, what is the meaning of this paragraph? And what does the ‘we’ refer to?
25. Line 584, “which has the NASA logo.” What’s the meaning of this statement?
26. The authors should expand the future part work. It is suggested to elaborate more about the future directions of PdM based on the authors review of recent achievements, which is important and expected in a review paper.
27. Figure 3, table 1 and 2 are not referenced in the text.
28. Grammar in line 132 to 133.
29. Overall, the authors should proofread the paper very carefully before submission. There are many grammar, wordings, and format issues. It is suggested that the authors reconsider the taxonomy of PdM techniques in Section 4.2 instead of using the categorization of specific ML algorithm as algorithms are various. The authors can consider more high-level taxonomy and specify the differences of various PdM technologies based on the authors judgement rather than directly using algorithm-level taxonomy. The taxonomy in table 2 and 3 are more clear and the authors should elaborate them and consider using these taxonomy criteria in Section 4.2.

Experimental design

comment are in 1. Basic reporting

Validity of the findings

comment are in 1. Basic reporting

Reviewer 2 ·

Basic reporting

This literature review on planning models and machine learning techniques for predictive maintenance is within the scope of the journal and of cross-disciplinary interest. In the Introduction, the topic and motivation of the study are clearly presented, being also the target audience correctly identified. Nevertheless, this field was already reviewed in 2019 (Carvalho et al. 2019), and the present work does not include substantial variations with respect to the previous one other than listing related papers published from November 2019 until the end of 2022. Neither does it approach the problem from a different perspective, nor does it provide an alternative taxonomy of the different machine learning tools.

As for the structure of the article, it is correct. However, I would suggest some changes, such as: (i) resorting to a table to provide a more visually friendly overview of the subset of papers reviewed (with columns such as topics, date of publication, machine learning techniques implemented, etc); and (ii) changing the structure of section 4 to correspond to the different branches and sub-branches of Machine Learning to which the different approaches belong.

Regarding English, it needs a thorough review since several sentences have subject-verb mismatches.

Experimental design

The description of the Survey Methodology is quite detailed but insufficient. I do not think that the study could be replicated with the information provided. This is because the authors state (lines 133-135) that “To compile the first list of publications for this research, significant sources including BUT NOT LIMITED TO Google Scholar, IEEE Xplore and Science Direct were surveyed”. Note that ALL sources and search keywords must be detailed so that the study can be replicated. Enumerating a subset of the sources is not enough. In this regard, I wonder why the authors did not search Scopus and Web of Science first. In addition, I infer that the second set of publications was extracted from the reference sections of the articles chosen in the first list. However, this is not clearly stated in the contribution. Furthermore, the authors separate the collected papers into ML for predictive maintenance and deep learning techniques for predictive maintenance with no justification or apparent reason for such segregation. As for the keywords selected to search the databases, the authors do not specify if the search is conducted with “AND”, “OR” or any other criteria. It is also remarkable that the term “Machine Learning” is not part of the keywords. From all the above, survey methods need to be revised. The authors should consider exploring Scopus and Web of Science as source databases, keywords may need to be expanded, and, eventually, all the process should be explained in greater detail, providing all the information necessary to ensure the replicability of the study.

As for the logical organization of the article into coherent paragraphs and subsections, the organization chosen is correct as it stands. Nonetheless, for the sake of clarity and to differentiate their contribution from previous literature reviews, I would recommend structuring the paper differently. In particular, I would structure section 4 around the three main branches of Machine Learning, namely: (i) Supervised Learning, (ii) Unsupervised Learning and (iii) Reinforcement Learning. More specifically, for each of them I would provide a paragraph with a succinct description of the branch, and I would enumerate the different ML techniques that belong to them (Regression and Classification to Supervised Learning, Clustering and Dimensionality Reduction to Unsupervised Learning, etc.). Once the different subfields of Machine Learning have been presented, I would keep this structure as the backbone of the subsequent sections of the article, i.e., I would present the different contributions according to the ML branch to which they belong.

Validity of the findings

The conclusions are clear and supported by the descriptions provided in previous sections. Nevertheless, note that if the authors include the suggestions made regarding the source databases, the keywords, and some of the comments that will be made in the next section (Additional Comments), these conclusions may change.

Additional comments

• In the Introduction, the section describing maintenance policy techniques is too long. I would summarize it in a single paragraph consisting of the enumeration of the different policies and a sentence for each of them. (This suggestion stems from the fact that only one technique is added in this section with respect to Carvalho et al. 2019, so it is no necessary to go into the detail on maintenance policies that can be consulted in the references).
• The definition of ML provided in lines 94-97 is vague and inaccurate. Please check ML introductory books or papers to refine the definition. See, for instance, An Introduction to Statistical Learning (https://www.statlearning.com/), or the paper by Rebala et al. 2019 (https://doi.org/10.1007/978-3-030-15729-6_1).
• In lines 100-101 it is important to clarify that performance depends not only on the ML technique selected, but also on the data available and the problem at hand.
• In section 3, when the authors describe the PdM planning model, I miss a reference to the literature (such as Abidi et al. 2022: https://www.mdpi.com/2071-1050/14/6/3387). In addition, it would be necessary to emphasize that the steps detailed are those of the PdM planning model, not mandatory steps for the general application of ML techniques to maintenance problems. For instance, even though data normalization and feature selection are steps of the PdM planning model, for the standalone application of ML techniques to maintenance problems they may or may not be necessary depending on the research question, the ML algorithms selected, the data available, etc.
• In section 3.2., line 176, the description of outliers is incorrect. The paragraph devoted to missing data and the different missing data imputation techniques is imprecise and incomplete as well. I recommend revisiting this section to correct it and to describe more rigorously what outliers and missing data consist of.
• Section 3.3. is also inaccurate, with the sentence in line 195 being totally incorrect. I recommend revising and correcting it in order to present the different normalization procedures more clearly.
• Section 3.4. needs improvement as well. There authors explain the feature selection procedure from the PdM planning model as if it were the only feature selection technique, which is misleading. They should first explain that different feature selection techniques exist and list them. Also, when they say that feature selection decreases attributes by half, I assume that this statement is linked to the PdM planning model. Notwithstanding, whatever it is, it needs to be supported by the appropriate references.
• In the Literature Survey section, the first remark is that it is no replicable. On the other hand, it is not comprehensive, as the terms “Machine Learning” or “Explainable AI” (XAI) have not been included in the keywords. Note as well that the search is restricted to some ML algorithms such as ANN, SVMs or RFs, excluding other techniques without adequate justification.
• Regarding Explainability, since the review is about ML, I miss that the authors state that most ML algorithms are black-box, i.e., difficult (if not impossible) to interpret. That is precisely the reason why XAI has emerged, to try to attain predictive accuracies similar to those of black-box algorithms while being interpretable. Note that explainability is particularly relevant in the context of industrial applications, where the different regressors typically represent physical variables on which we can act. Notably, different works linking explainable machine learning to predictive maintenance already exist in the literature (Vollert et al. 2022, Hrnjica & Softic 2021).
• In 4.2.4. Fault Diagnostics and Anomaly detection are presented as other AI approaches, when they constitute alternative research questions that are also addressed with some of the ML tools described above.
• Eventually, in the Conclusions section, the authors talk about ML algorithms such as GBM or XGBoost that have not been mentioned or explained earlier in the paper. These techniques cannot appear for the first time in the conclusions.

---

## Round 0.2 · Minor Revisions

Dear Authors,

After careful consideration of your manuscript titled "Predictive maintenance in industry 4.0: A survey of planning models and machine learning techniques" and the feedback from our reviewers, we have concluded that your manuscript may be acceptable for publication in PeerJ Computer Science following minor revisions.

The reviewers appreciate your efforts to address previous suggestions but have identified areas needing further refinement. These include enhancing the abstract for a more comprehensive summary, ensuring consistency in language use, improving the clarity and detail in your methodology for replicability, and refining sections on machine learning to align with the theme of predictive maintenance. Additionally, attention to detail in the use of acronyms and the inclusion of relevant references, particularly in emerging areas like Explainable AI, is recommended to maintain the relevance and accuracy of your review.

Please address these overarching concerns in your revised manuscript. We look forward to receiving your updated submission and thank you for considering PeerJ Computer Science for your work.

Sincerely,

José M. Galán

Reviewer 1 ·

Basic reporting

1. The abstract needs rework. The abstract needs to summarize the problem you are studying in this paper, the motivation and necessity of doing this literature review, a brief intro of your review (include the analysis method, and findings/results), the significance of this work, vision and future thoughts. The current version of Abstract poorly includes the above points.
2. References shouldn’t be added for sections/subsections titles.
3. Part of the section 3.3 is related to feature selection. The authors are suggested to think if this part should be put in section 3.4.
4. Section 3.5.1 to 3.5.5 need some references for each type of technology.
5. Decision tree is suggested to be introduced before random forest since RF is an ensemble of multiple DTs.
6. The authors reworked section 4.2 to include more literatures, which is good. However, some of them are irrelevant to predictive maintenance applications but for ML model illustration purpose. It's better to focus on the theme of this paper. Maybe the authors can keep some of them while concentrating on the theme of this paper.

Experimental design

no comment

Validity of the findings

no comment

Reviewer 2 ·

Basic reporting

The authors have thoroughly addressed most suggestions made by the reviewers. Nevertheless, some issues remain open and need to be tackled.
Regarding English, a new review would be necessary since the manuscript still has sentences with subject-verb mismatches. In addition, the authors use both British English and American English within the whole document. They should choose one of them and adhere to its grammar, but not mix the two.

Experimental design

The Survey Methodology has been described in greater detail, but it is still insufficient. The authors continue to not specify whether the search is conducted with “AND”, “OR” or any other criteria, which would be necessary to ensure the replicability of the study.

Validity of the findings

The conclusions are clear and supported by the descriptions provided in previous sections.

Additional comments

• Regarding acronyms, when introducing them for the first time, it is essential to specify both the initial letters that constitute the acronym and the words they represent. Nevertheless, in subsequent paragraphs, only acronyms should be used. It is not correct to continue specifying both the words and the initials throughout the entire document.
• The definition of ML, which was initially vague, now is absent. A general definition of ML, to be found in introductory books or papers, would be needed. See, for instance, An Introduction to Statistical Learning (https://www.statlearning.com/), or the paper by Rebala et al. 2019 (https://doi.org/10.1007/978-3-030-15729-6_1).
• Including missing data imputation techniques within the data cleansing section is misleading. It should not be there, and it is important to state that it is not always necessary to impute missing data for advanced data analysis in industrial contexts.
• The authors systematically distinguish between machine learning and deep learning, despite deep learning being a subfield of machine learning. It would be necessary to provide justification for why such differentiation is made. Otherwise, they should simply refer to it as machine learning.
• The keywords list has been enlarged to include terms such as “Machine Learning”. However, the authors have disregarded including “Explainable AI” (XAI). XAI has been left as a future research line. I do not agree with this decision, because it translates into the review becoming both incomplete and a bit outdated. Note that many of the most recent advancements in machine learning for industrial applications are linked to XAI —see for instance (Cummins et al. 2024, Vollert et al. 2022, Hrnjica & Softic 2021).

Annotated reviews are not available for download in order to protect the identity of reviewers who chose to remain anonymous.

---

## Round 0.3 · accepted · Accept

I believe that the proposed changes have been adequately corrected by the authors and the article is now suitable for publication. Congratulations